A taxonomic revision of the Sinopterus complex (Pterosauria, Tapejaridae) from the Early Cretaceous Jehol Biota, with the new genus Huaxiadraco

http://orcid.org/0000-0003-2909-3346 Pêgas Rodrigo V. 1
http://orcid.org/0000-0003-1368-9831 Zhou Xuanyu 2 3 4 xyzhou@elms.hokudai.ac.jp
Jin Xingsheng 5
Wang Kai 6
http://orcid.org/0000-0002-6095-9112 Ma Waisum 7
1 Laboratório de Paleontologia de Vertebrados e Comportamento Animal, Universidade Federal do ABC , São Bernardo do Campo , Brazil
2 Department of Natural History Sciences, Hokkaido University , Sapporo, Hokkaido , Japan
3 Hokkaido University Museum, Hokkaido University , Sapporo, Hokkaido , Japan
4 Beipiao Pterosaur Museum of China , Beipiao, Liaoning , China
5 Zhejiang Museum of Natural History , Hangzhou, Zhejiang , China
6 School of Earth Sciences and Resources, China University of Geosciences , Beijing , China
7 Department of Paleobiology, National Museum of Natural History, Smithsonian Institution , Washington D.C. , United States
Young Mark
Electronic publication date: 2023 Feb 9
Publication date: 2023
Volume: 11
Electronic Location ID: e14829
Received 2022 Nov 3; Accepted 2023 Jan 9
Copyright: © 2023 Pêgas et al.
Copyright year: 2023
Copyright holder: Pêgas et al.
License: This is an open access article distributed under the terms of the Creative Commons Attribution License, which permits unrestricted use, distribution, reproduction and adaptation in any medium and for any purpose provided that it is properly attributed. For attribution, the original author(s), title, publication source (PeerJ) and either DOI or URL of the article must be cited.
License URL: https://creativecommons.org/licenses/by/4.0/

Keywords: Taxonomy, Pterosaur, Linear morphometrics, Ontogeny, Allometry, Phylogeny, Jiufotang Formation, Osteology

Funding: FAPESP #2019/10231-6 Hokkaido University DX Doctoral Fellowship #JPMJSP2119 National Museum of Natural History Smithsonian Institution Key Laboratory of Stratigraphy and Palaeontology Ministry of Natural Resources #KLSP2101 This work was supported by FAPESP (#2019/10231-6), the Hokkaido University DX Doctoral Fellowship (#JPMJSP2119), the Deep Time Peter Buck Postdoctoral Fellowship from National Museum of Natural History, Smithsonian Institution, the Fund from the Key Laboratory of Stratigraphy and Palaeontology, Ministry of Natural Resources (#KLSP2101). The funders had no role in study design, data collection and analysis, decision to publish, or preparation of the manuscript.

==============================
Tapejarids are edentulous pterosaurs particularly abundant in the Chinese Jiufotang Formation, counting with over 10 described specimens and dozens of undescribed ones. A total of seven nominal tapejarid species (within two genera) have been proposed, though it is disputed how many of those are valid instead of sexual or ontogenetic morphs of fewer, or a single, species. However, detailed revisions of the matter are still lacking. In the present work, we provide a specimen-level survey of anatomical variation in previously described Jiufotang tapejarid specimens, as well as of six new ones. We present qualitative and morphometric comparisons, aiming to provide a basis for a taxonomic reappraisal of the complex. Our results lead us to interpret two Jiufotang tapejarid species as valid: Sinopterus dongi and Huaxiadraco corollatus (gen. et comb. nov.). Our primary taxonomic decisions did not rely around cranial crest features, which have typically been regarded as diagnostic for most of these proposed species albeit ever-growing evidence that these structures are highly variable in pterosaurs, due to ontogeny and sexual dimorphism. However, a reassessment of premaxillary crest variation in the Sinopterus complex reveals that while much of the observed variation (crest presence and size) can easily be attributed to intraspecific (ontogenetic and sexual) variation, some of it (crest shape) does seem to represent interspecific variation indeed. A phylogenetic analysis including the species regarded as valid was also performed.

Introduction

The Tapejaridae (sensu Andres, 2021) are a clade of Cretaceous edentulous pterosaurs of the group Azhdarchoidea (Pterodactyloidea, Eupterodactyloidea), characterized by their short, downturned rostra and peculiar premaxillary crests (Kellner & Campos, 2007; Pêgas, Leal & Kellner, 2016). They comprise over 10 species (up to 14 valid species following Zhang et al., 2019), spanning from the Barremian to the Santonian; with records from Brazil, Morocco, Europe, and China (Kellner & Campos, 2007; Vullo et al., 2012; Andres, Clark & Xu, 2014; Pêgas, Leal & Kellner, 2016).

Tapejarids are a relatively common element of the famous Jehol Biota of China. From the Yixian Formation, a single species has been described: Eopteranodon lii, represented by two specimens (Lü & Zhang, 2005; Lü et al., 2006c). Originally regarded as an undetermined pterodactyloid (Lü & Zhang, 2005) or a pteranodontid (Lü et al., 2006c), it was later reinterpreted as a tapejarid (Andres & Ji, 2008; Vullo et al., 2012). In contrast with the Yixian Fm. (late Barremianearly Aptian), a great abundance of tapejarids is found in the Jiufotang Formation (Aptian). In total, 15 specimens of Jehol tapejarids have been formally described in the literature (Wang & Zhou, 2003; Li, Lü & Zhang, 2003; Lü & Zhang, 2005; Lü & Yuan, 2005; Lü et al., 2006a, 2006b, 2006c, 2007, 2016; Liu et al., 2014; Zhang et al., 2019; Shen et al., 2021; Zhou, Niu & Yu, 2022; Zhou et al., 2022). Under the accounts of Shen et al. (2021), the total number of recovered specimens, scattered around Chinese institutions, must be close to a hundred.

The first tapejarid to be recovered from China was Sinopterus dongi, from the Jiufotang Formation (see Wang & Zhou, 2003). Further six Jiufotang tapejarid species have been named posteriorly: Sinopterus gui, Sinopterus lingyuanensis, Huaxiapterus jii, Huaxiapterus corollatus, Huaxiapterus benxiensis, and Huaxiapterus atavismus (see (Wang & Zhou, 2003; Li, Lü & Zhang, 2003; Lü & Yuan, 2005; Lü et al., 2006a, 2007, 2016). These proposed speices of Jiufotang tapejarids are involved in a series of complex taxonomic disputes, with the genera Huaxiapterus and Sinopterus having been synonymized (Wang & Zhou, 2006; Wang et al., 2008; Witton, 2013; Zhang et al., 2019). Thus, the Jiufotang tapejarids will heretofore be referred to as the Sinopterus complex.

The type species Sinopterus dongi was described by Wang & Zhou (2003) and its validity has never been contested. A second species, Sinopterus gui, was proposed by Li, Lü & Zhang (2003), but its holotype was later reinterpreted as an undiagnostic juvenile specimen, indistinct from S. dongi (Kellner & Campos, 2007; Kellner, 2010).

Following the description of these two species, the genus Huaxiapterus was erected for the type-species Huaxiapterus jii by Lü et al. (2005). Afterwards, Wang & Zhou (2006) synonymized Huaxiapterus jii with Sinopterus dongi, regarding that the two holotypic specimens were indistinguishable. Kellner & Campos (2007) accepted the validity of H. jii at the species level, but referred it to the genus Sinopterus, as Sinopterus jii. Later, however, Kellner (2010) and Zhang et al. (2019) regarded S. jii as a synonym of S. dongi, following the proposition by Wang & Zhou (2006). A consequence of this species-level synonymy is that the genus Huaxiapterus would become invalid.

Later, Lü et al. (2006a) attributed a second species to the genus Huaxiapterus, H. corollatus. Kellner & Campos (2007) accepted the species-level validity of H. corollatus and suggested that it required a new genus name (recognizing the proposed synonymy between H. jii and S. dongi, and considering that H. corollatus was sufficiently distinct from S. dongi to warrant another genus name). Later, another species was proposed for the genus Huaxiapterus by Lü et al. (2007): Huaxiapterus benxiensis.

Subsequently, Witton (2013) proposed that the majority of the previously described Jiufotang tapejarids could possibly represent a single ontogenetic continuum. Witton (2013) noticed that the diagnoses of the proposed species relied heavily on crest size and shape, what is problematic since this is most likely strongly influenced by sexual and ontogenetic variation (e.g., Bennett, 1993; Wang et al., 2014; Manzig et al., 2014; Pinheiro & Rodrigues, 2017). Though Witton (2013) made a case for this possibility, it has never been investigated in detail so far. Andres, Clark & Xu (2014) did not contest the validity of any of the previously proposed species, having coded all the then-described species in their phylogenetic analysis: Sinopterus dongi, Huaxiapterus jii, Sinopterus gui, Huaxiapterus corollatus and Huaxiapterus benxiensis.

More recently, Lü et al. (2016) rejected all proposed synonymies and further proposed two new species, Sinopterus lingyuanensis and Huaxiapterus atavismus. Subsequently, Zhang et al. (2019) sank all species ever attributed to Huaxipterus onto Sinopterus, and recognized five species as valid: Sinopterus dongi, Sinopterus corollatus, Sinopterus benxiensis, Sinopterus lyngyuanensis and Sinopterus atavismus. Zhang et al. (2019) regarded Sinopterus gui and Sinopterus jii as junior synonyms of Sinopterus dongi. Still, Zhang et al. (2019) did not present detailed discussions concerning this taxonomic proposal.

Subsequently, Naish, Witton & Martin-Silverstone (2021) preliminarily corroborated the proposition of Witton (2013) that all Jiufotang tapejarids represent an ontogenetic continuum of a single species. Still, Naish, Witton & Martin-Silverstone (2021) noted that at least Huaxiapterus corollatus was an apparent outlier regarding limb proportions, thus suggesting that it “may represent a second taxon”, pending further testing. More recently, Shen et al. (2021) supported the proposition by Naish, Witton & Martin-Silverstone (2021).

In summary, a total of seven tapejarid species have been proposed for the Jiufotang Formation, all eventually attributed to the genus Sinopterus and intricated in a series of complex disputes based on preliminary considerations. A detailed review of the Sinopterus complex is still lacking, and a critical survey of anatomical variation is thus of the uttermost importance. The present work aims at: A specimen-level assessment of morphological variation within the Sinopterus complex. For this, we present qualitative anatomical comparisons (specimen by specimen) as well as quantitative analyses (allometric and linear morphometric analyses), englobing previously described specimens as well as six new specimens;

An interpretation of the surveyed variation (as either intra- or interspecific), in order to interpret the validity and circumscription of each species. Our primary delimitation of species will disregard cranial crest features. After our primary delimitation has been made, we will proceed to map cranial crest variation and interpret it.

Inferring the phylogenetic relationships between the established valid species.

With these considerations, we hope to reinterpret the Sinopterus complex and provide a taxonomic reassessment, based on which new specimens can be identified. Pivotal to the taxonomic history of the Sinopterus complex is the role of cranial crests in pterosaur taxonomy. It is clear that cranial crest features used alone make for problematic taxonomic decisions (Witton, 2013), as they could rather reflect ontogenetic or sexual variations (Bennett, 1993; Wang et al., 2014; Manzig et al., 2014; Pinheiro & Rodrigues, 2017). However, it is also clear that some closely related species may exhibit disparate cranial crest morphologies (at least when inferred mature males are considered), which can thus contain taxonomic signal (e.g., Pteranodon longiceps and Pteranodon sternbergi; see Bennett, 1994). It is for this reason that, in this work, we aim at revising the taxonomy of the Sinopterus complex with extra caution regarding cranial crest variation, by making a primary taxonomic assessment without input from cranial crest data first, and then assessing and interpreting cranial crest variation subsequently; instead of using cranial crest variation as an a priori source of taxonomic signal.

Material and Methods

Geological setting

The Jiufotang Formation is widely distributed in the terrestrial volcanic sedimentary basins of northern Hebei and western Liaoning, which have yielded the diverse Jehol Biota (Xi et al., 2019). It represented a lacustrine environment surrounded by temperate forests (Zhou, Barrett & Hilton, 2003; Benton et al., 2008). Although specimens are typically crushed, preservation is nonetheless exceptional and soft tissue is often found (Benton et al., 2008; Zhou & Wang, 2010).

The Jiufotang Formation of western Liaoning is distributed within six continental faulted basins, trending northeast: Fuxin-Yixian Basin, Beipiao-Chaoyang Basin, Dapingfang-Meileyingzi Basin, Dachengzi-Siguanyingzi Basin, Jianchang Basin, Lingyuan-Sanshijiazi Basin (Su et al., 2008; Wu et al., 2018; Xi et al., 2019).

The rock layers are mainly grey to greyish green in color, interbedded with greyish yellow, greyish white, greyish black and occasionally purple rocks (Wu et al., 2018). They consist of calcareous silty shales, shales, and siltstones, interbedded with oil shales, tuffs, bentonites, coal seams, marlstones, sandstones, and conglomerates (Wu et al., 2018). This sedimentary association is dominated by lake sediments and includes abundant macrofossils of animals and plants. The thickness of Jiufotang Formation varies from ~200–3,000 m depending on locality, contacting the underlying Yixian Formation through a parallel unconformity (Wu et al., 2018). It is overlaid by formations as among which Binggou Formation and Fuxin Formation.

Unique fossil-bearing bed (UFBB) refers to a set of Chinese national key protected fossils (classified as level three or above, by National Standard for classification of Paleontological Fossils, China), such as reptiles and birds, which is known from a regionally stable and significant geological formation. A number of unique fossil-bearing beds have been named (e.g., Duan et al., 2006, 2010; Wu et al., 2018, Gao et al., 2018).

Wu et al. (2018) divided the Jiufotang Formation into three sections from bottom to top, based on lithology, depositional cycle, basic sequence, and fossil assemblage. In general, the base of every section consists of yellowish brown-yellowish green, thick-bedded medium to coarse conglomerate. The top layer is made up of yellowish green thin to very thin tuffaceous siltstone and thin silty mudstone. A short-term cycle is formed by conglomerate (containing glutenite), sandstone, siltstone and shale. About seven to nine short-term cycles form a mid-term cycle (three sections of Jiufotang Formation) that exhibits finer grain sizes and thinner beds progressively upward, as shown in a schematic division and correlation diagram of the Jiufotang Formation and the UFBB in western Liaoning (including five basins: Fuxin-Yixian Basin, Beipiao-Chaoyang Basin, Dapingfang-Meileyingzi Basin, Dachengzi-Siguanyingzi Basin and Jianchang Basin; Wu et al., 2018). Detailed paleoenvironmental reconsctructions for each bed are still needed.

Due to the highly fossiliferous nature of the Jehol Group, several fossils are commonly found by local collectors, although without a precise control over their stratigraphic provenance (e.g., Kellner, 2010; Lü et al., 2016). A notable exception is the holotype of Sinopterus dongi, known to come from the Lamagou UFBB, of the Second Member of the Jiufotang Formation (Zhang et al., 2007). The holotypes of Sinopterus gui and Huaxiapterus jii come from the mudstone/shale layers of Nanlu, Shengli Town, which correspond to the Yuanjiawa UFBB of the Third Member of the Jiufotang Formation (Zhang et al., 2007). Specimens PMOL-AP00030 and D3072 are known to have come from the Dapingfang locality (Liu et al. 2014; Shen et al., 2021), where the Third Member of the Jiufotang Fm. outcrops (see Wu et al., 2018).

The holotype of Sinopterus lingyuanensus and specimen IVPP V 23388 are known to come from Sihedang, Lingyuan, and they are preserved in shales (Lü et al., 2016; Zhang et al., 2019), what indicates they likely come from the Third Member Sihedang beds (see Wu et al., 2018).

The holotype of Huaxiapterus benxiensis is reported to come from Lianhe Town (Lü et al., 2007), and thus from the Dapingfang Basin, meaning it comes from either the Second or Third Member (Zhang et al., 2007; Wu et al., 2018).

For specimens D2525 and the holotype of H. corollatus, the only information available is that they come from Chaoyang City (Lü et al., 2006b, 2007). The same applies to the new specimens reported here (D4019, BPMC 103, BPMC 104, BPMC 105, BPMC 106, and BPMC 107). Within Chaoyang City, two fossiliferous beds of the Jiufotang Formation occur: the Dongpochi Bed of the Second Member, and the Shangheshou Bed of the Third Member (Zhang et al., 2007). Unfortunately, it is hard to define from which bed came each of the remaining Jiufotang tapejarid specimens, but it can be said that they come from either the Second or the Third Member.

Morphometric dataset

We have compiled a morphometric dataset for the purposes of our allometric and morphometric clustering analyses. We have coded in our dataset 14 relatively complete Jiufotang tapejarid specimens: the holotypes of Sinopterus dongi (IVPP V 13363, Wang & Zhou, 2003), Huaxiapterus jii (GMN-03-11-001, Lü et al., 2005), Huaxiapterus corollatus (ZMNH M813, Lü et al., 2006a), Huaxiapterus benxiensis (BXGM V0011, Lü et al., 2007), Sinopterus lingyuanensis (JPM-2014-005, Lü et al., 2016) and Huaxiapterus atavismus (XHPM 1009, Lü et al., 2016), as well as specimens D2525 (Lü et al., 2006b), IVPP V 23388 (Zhang et al., 2019), D3702 (Shen et al., 2021), and five new specimens which are presented here for the first time (D4019, BPMC 103, BPMC 104, BPMC 105, and BPMC 107). The holotype of Sinopterus gui (see Li, Lü & Zhang, 2003), specimens PMOL-AP00030 (Liu et al. 2014), SDUST-V1012 (Zhou, Niu & Yu, 2022), and SDUST-V1014 (Zhou et al., 2022), as well the new specimen BPMC 106, were not included in the morphometric dataset due to their high level of incompleteness. The holotype of Nemicolopterus crypticus is also not included due its extremely young age (most likely a near-hatchling; Naish, Witton & Martin-Silverstone, 2021) and the incompleteness of its wings (Wang et al., 2008).

For comparative purposes, we have also included in our morphometric analyses other tapejarid species, namely Eopteranodon lii, Tapejara wellnhoferi, Caiuajara dobruskii and Tupandactylus navigans. We have included the two known specimens of Eopteranodon lii, which come from the Yixian Formation: the holotype BPV-078 (Lü & Zhang, 2005), and the referred specimen D2526 (Lü et al., 2006c). Our entry for Tapejara wellnhoferi is based on specimens SMNK PAL 1137 (Eck, Elgin & Frey, 2011) and AMNH 24445 (Vila Nova et al., 2015). Tupandactylus navigans is based on specimen GP/2E 9266 (Beccari et al., 2021), and Caiuajara dobruskii is based on a combination of specimens CP.V 872a, CP.V 1006, and CP.V 1001b (Manzig et al., 2014). Data for Caiuajara specimens was taken from Manzig et al. (2014). Data for all other specimens was taken first-hand.

We compiled a morphometric dataset of 21 skeletal measurements, among six skull measurements and 15 postcranial elements. The analyzed skull measurements comprise rostral index, rostral value, rostrum deflection angle, length/height ratio of the nasoantorbital fenestra, orbit ventral angle, and quadrate reclination angle. The postcranial bone measurements comprise the length of the fourth cervical, fifth cervical, humerus, ulna, metacarpal I, metacarpal IV, wing phalanges 1–4, femur, tibia, metatarsal I, and metatarsal II. A spreadsheet containing our morphometric dataset is available as File S1 (Sheet 1).

Allometric correlation analysis

The dataset for the correlation and allometric analyses was restricted to specimens of the Sinopterinae, more specifically the Jehol tapejarids (Jiufotang tapejarids plus Eopteranodon lii from the Yixian Fm.), which are deemed as a complex of closely related and rather conservative species, and thus similar ontogenetic trends were assumed. The same assumption cannot be made for more distantly related tapejarid taxa, which were thus left aside from these analyses in order to avoid potential noise. A spreadsheet containing our dataset for the allometric correlations (log-transformed morphometric values for the Sinopterinae only) is available within our File S1 (Sheet 2).

Bivariate allometric analyses were performed to test for correlation to size variation and potential allometric relationships. We utilized the standardized major axis (SMA) line-fitting method to determine the allometric equation (Warton et al., 2006), largely following the protocol of Yang et al. (2022). We utilized humeral length as the common independent variable (i.e., as a proxy for body size), that is, using it as the common parameter for assessing morphometric variables in different specimens, and thus aiming at testing potential correlations and allometric relationships between morphometric variables and body size. All values were log-transformed for the SMA analyses.

We thus performed the SMA analyses between log-transformed values of humeral length and each of the analyzed proportions: rostral index, rostral value, rostrum deflection angle, length/height ratio of the nasoantorbital fenestra, orbit ventral angle, quadrate reclination angle, fourth cervical length, fifth cervical length, ulna length, metacarpal I length, metacarpal IV length, wing phalanges 1–4 lengths, femur length, tibia length, metatarsal I length, and metatarsal II length. The p-value was calculated in order to test for correlation between body size variation (as indicate by humeral length as a proxy) and each analyzed variable. For each variable, if the correlation was statistically significant (p < 0.05), then the allometric correlation was performed for this variable. If the correlation was not statistically significant (i.e., a certain variable does not correlate to body size), then the variable in question can be interpreted as not ontogenetically variable, and thus allometry is not calculated for this variable. The SMA analyses were undertaken using the software PAST (Hammer, Harper & Ryan, 2001). Isometry is considered as the null hypothesis.

Typically, a correlation is deemed as isometric when, in the line fitting equation, slope equals (or is insignificantly different from) 1 (see Warton et al., 2006; Yang et al., 2022). In contrast, the correlation is deemed as negatively allometric and positively allometric when slope is, respectively, significantly lower and higher than 1. To determine this, 95% confident intervals (with 1,000 iterations) were calculated for the slope for each SMA analysis. The null hypothesis (isometry) is rejected if the slope’s 95% confidence interval (CI) lays entirely above or below 1, indicating, respectively, positive or negative allometry. If the CI is comprised between a lower value below 1 and an upper value above 1 (i.e., 1 is comprised within the CI), then isometry is assumed.

Linear morphometric multivariate analyses

After our SMA analyses, we constructed a morphometric dataset of skeletal proportions (all log-transformed) based on skeletal elements devoid of allometric signal, as per the results of the previous SMA analyses. Thus, aiming at excluding noise from data that is allometrically correlated to size variation, our morphometric dataset aims to be based on two types of morphometric data: (1) data that is not correlated to size variation and (2) data that is isometrically correlated to size variation. Afterwards, our pruned tapejarid morphometric dataset (including all tapejarid taxa) was subjected to an unweighted pair-group average (UPGMA) cluster analysis (using Euclidean distance) as well as a Principal Component Analysis (PCA). Two different PCA analyses were performed, each differing in the treatment of missing data: one using mean values imputation, and one using iterative imputation. These analyses were executed using the software PAST (Hammer, Harper & Ryan, 2001). A spreadsheet containing our dataset for the multivariate analyses (preened skeletal angles and proportions for all analyzed tapejarids) is available within our File S1 (Sheet 3).

Non-parametric tests

After the performance of the multivariate analyses as described above, the analyzed sample of Jiufotang tapejarids was divided into two separate groups (morphotypes), as expressed further below in the Results section. For the purpose of testing the significance (or lack thereof) of the difference between the proposed groups regarding each morphometric value, the non-parametric test of Kruskal-Wallis was performed. Kruskal-Wallis was performed for all analyzed morphometric values for which n > 2 for each of the two groups (morphotypes). Source-data was log-transformed, as for the allometric and multivariate analyses. The level of significance (alpha-value) was set at 0.05. A spreadsheet containing our dataset for the Kruskal-Wallis analyses (per variable) is available within our File S1 (Sheet 4). The analyses were also carried out using the software PAST (Hammer, Harper & Ryan, 2001).

Ontogenetic assessment

For the purpose of assessing the ontogenetic stages of the studied specimens, we follow here the many criteria put forward by workers such as Bennett (1993), Kellner & Tomida (2000), and Kellner (2015). However, we do not strictly follow the “five ontogenetic stages” model based on bone fusion sequence (Kellner, 2015), since not all pterosaur clades exhibited similar sequences of ontogenetic bone fusion (Dalla Vecchia, 2018). For the purpose of a relative assessment of ontogenetic development within the Sinopterus complex, the specimens are here compared to each other only (based on bone fusion), and thus put in a restricted, in-clade context (File S1, Sheet 6). For body size context within the ontogenetic assessment, a scatter plot of humerus length/maximized wingspan is provided (File S1, Sheet 6). Maximized wingspan corresponds to the absolute sum of coracoid, humerus, ulna, metacarpal IV, and wing finger (e.g., Kellner et al., 2013). For incomplete specimens in which a given element is missing (see File S1, Sheet 1), the missing element was estimated based on the mean value of the proportion between the element in question and the humerus according to the rest of the sample.

Phylogenetic analysis

Subsequent to our reassessment of the species-level taxonomy of the Sinopterus complex, we proceeded to perform a phylogenetic analysis, which is the last step of the present work. After obtaining the results from our taxonomic reassessments (see below for our taxonomic proposals and species circumscriptions), we included and coded all Chinese tapejarid species (those that were considered as valid here) in an updated version of the data matrix from Pêgas et al. (2021). For this reason, in the present article, a separate Phylogenetic Analysis section is presented only after the main Discussion section.

We performed a cladistic analysis using the software TNT 1.5 (Goloboff, Farris & Nixon, 2008), which was divided in two steps, following the same protocol as previously described by Wei et al. (2021). New Technology Search was used for the first search (using Sectorial Search, Ratchet, Drift and Tree fusing, default parameters), with random seed = 0. In sequence, using trees from RAM, a traditional search swapping was performed (using TBR, 10,000 replications, collapsing trees after search). All characters were treated with equal weights. A Mesquite file (Nexus format) containing the data matrix is available as File S2. A TNT file, ready for analysis execution in TNT, is available as File S3.

Coding for Bakonydraco galaczi is restricted to jaw elements (Ősi, Weishampel & Jianu, 2005; Ősi, Buffetaut & Prondvai, 2011). Coding for Afrotapejara zouhri is based on the holotype and the three referred specimens (Martill et al., 2020a). Coding for Aerotitan sudamericanus follows the interpretation of the holotype as a lower jaw (Pêgas et al., 2021; contra Andres, 2021). The holotype of Alanqa saharica is also coded here as a lower jaw (Pêgas et al., 2021; contra Ibrahim et al., 2020); however, its coding is corrected here based on an anatomical reinterpretation, with a dentary occlusal eminence being absent and instead a pair of dentary raised ridges being present (R. Smith & D. Martill, 2022, personal communication; see also Ibrahim et al., 2020), similar to that seen in specimen FSAC KK 4000 (Martill & Ibrahim, 2015; Ibrahim et al., 2020).

Nomenclatural acts

The electronic version of this article in Portable Document Format (PDF) will represent a published work according to the International Commission on Zoological Nomenclature (ICZN), and hence the new names contained in the electronic version are effectively published under that Code from the electronic edition alone. This published work and the nomenclatural acts it contains have been registered in ZooBank, the online registration system for the ICZN. The ZooBank LSIDs (Life Science Identifiers) can be resolved and the associated information viewed through any standard web browser by appending the LSID to the prefix http://zoobank.org/. The LSID for this publication is: urn:lsid:zoobank.org:pub:E836D564-B986-497A-9E3C-8277EF8EF50E. LSID for the new genus: urn:lsid:zoobank.org:act:39AA06E5-6882-4041-9585-8F2106424C81.

Phylogenetic nomenclature

The present work favors the recent propositions of the PhyloCode (de Queiroz, Cantino & Gauthier, 2020) as a means of standardizing and stabilizing phylogenetic nomenclature. We thus primarily follow the phylogenetic definitions given and registered by Andres (2021) and Pêgas et al. (2021) concerning azhdarchoids, though with a few unrestricted emendations. The phylogenetic nomenclatural scheme employed here, following recommendations of the PhyloCode, is presented in Table 1.

Table 1 Systematic nomenclature.

Clade	Nominal author	Definition	Composition and remarks	ICPN conversion and Regnum code	
Tapejaroidea	Kellner (2003)	The least inclusive clade containing Tapejara wellnhoferi Kellner, 1989, Quetzalcoatlus northropi Lawson 1975, and Dsungaripterus weii Young 1964.	Includes the sister-taxa Dsungaripteridae and Azhdarchoidea.	This work, [820].	
Azhdarchoidea	Unwin (1995)	The least inclusive clade containing Tapejara wellnhoferi Kellner, 1989 and Quetzalcoatlus northropi Lawson 1975.	Includes the sister-taxa Tapejaromorpha and Azhdarchomorpha.	Andres (2021), [355].	
Tapejaromorpha	Andres, Clark & Xu (2014)	The most inclusive clade containing Tapejara wellnhoferi Kellner, 1989 but not Azhdarcho lancicollis Nessov 1984.	Includes the sister-taxa Tapejaridae and Thalassodromidae.	Andres (2021), [356].	
Thalassodromidae	Witton (2009)	The least inclusive clade containing Thalassodromeus sethi Kellner
& Campos 2002 and Tupuxuara longicristatus Kellner & Campos 1988.	Includes Thalassodromeus, Tupuxuara, and Kariridraco.	Andres (2021), [770].	
Tapejaridae	Kellner (1989)	The least inclusive clade containing Tapejara wellnhoferi Kellner, 1989, Sinopterus dongi Wang & Zhou, 2003, and Caupedactylus ybaka Kellner, 2013.	The first registered definition (Andres, 2021) is (unrestrictedly) emended here in order to stabilize the clade’s diagnosis, usage, and content, under the context of the present reference phylogeny. Characterized mainly by downturned rostra and tall rostral crests, it contains Caupedactylia and Eutapejaria.	Andres (2021), [357], unrestrictedly emended here.	
Caupedactylia	This work.	The most inclusive clade containing Caupedactylus ybaka Kellner, 2013 but not Tapejara wellnhoferi Kellner, 1989.	Includes Caupedactylus and Aymberedactylus. This clade contains tapejarids which share a symphyseal shelf dorsoventrally steep and deep, and a flat dentary fossa.	This work,
[821].	
Eutapejaria	This work.	The most inclusive clade containing Tapejara wellnhoferi Kellner, 1989 but not Caupedactylus ybaka Kellner, 2013.	This clade contains tapejarids which share a dorsal dentary eminence, encompassing Tapejarinae and Sinopterinae (sensu Andres, 2021).	This work,
[822].	
Azhdarchomorpha	Pêgas et al. (2021)	The most inclusive clade containing Azhdarcho lancicollis Nessov 1984 but not Thalassodromeus sethi Kellner & Campos 2002 or Tapejara wellnhoferi Kellner, 1989.	Includes Keresdrakon, Chaoyangopteridae, Alanqidae, and Azhdarchidae.	Pêgas et al. (2021), [574].	
Chaoyangopteridae	Lü et al. (2008)	The most inclusive clade containing Chaoyangopterus zhangi Wang & Zhou, 2003 but not Quetzalcoatlus northropi Lawson 1975.	Includes Chaoyangopterus, Jidapterus, Shenzhoupterus, and Lacusovagus.	Andres (2021), [368].	
Azhdarchiformes	Andres (2021)	The most inclusive clade containing Quetzalcoatlus northropi Lawson 1975 but not Chaoyangopterus zhangi Wang & Zhou, 2003.	Under the present reference phylogeny, the Azhdarchiformes include Alanqidae and Azhdarchidae.	Andres (2021), [771].	
Alanqidae	Pêgas et al. (2021)	The most inclusive clade containing Alanqa
saharica Ibrahim et al. 2010 but not Chaoyangopterus zhangi Wang & Zhou, 2003 or Azhdarcho lancicollis Nessov 1984.	Includes Alanqa, Argentinadraco, Xericeps, Leptostomia, and Montanazhdarcho. Characterized by bowed-out lateral jaw margins in cross-section, and possibly by a pair of dentary occlusal ridges.	Pêgas et al. (2021), [576].	
Azhdarchidae	Padian (1986)	The least inclusive clade containing Azhdarcho lancicollis Nessov 1984, Phosphatodraco mauritanicus Pereda-Suberbiola et al. 2003, and Quetzalcoatlus northropi Lawson 1975.	Includes Eurazhdarcho, Aralazhdarcho, Phosphatodraco, Wellnhopterus, Zhejiangopterus, Azhdarcho, and Quetzalcoatlinae. Characterized by a vestigial cervical neural spine.	Andres (2021), [371]. Emended by Pêgas et al. (2021).	
Note:

Reference phylogeny: this work.

Of particular note concerning phylogenetic nomenclature in azhdarchoids is the conflicting usages of the terms Tapejaridae, Tapejarinae, and Thalassodrominae. Originally, the family Tapejaridae was erected in order to encompass Tapejara wellnhoferi and Tupuxuara longicristatus (Kellner, 1989), and later defined as the least inclusive clade containing these two taxa (Kellner, 2003). Tapejaridae was later divided into Tapejarinae and Thalassodrominae, which can be roughly described, respectively, as a “Tapejara-Sinopterus group” and a “Thalassodromeus-Tupuxuara group” (Kellner & Campos, 2007). Disagreement over the sister-group relationship between the “Tapejara-Sinopterus group” and the “Thalassodromeus-Tupuxuara group” led to a restrictive redefinition of the Tapejaridae by some workers, as the least inclusive clade containing Tapejara wellnhoferi and Sinopterus dongi, with the “Thalassodromeus-Tupuxuara group” thus elevated to a family-level Thalassodromidae (Lü et al., 2006a; Andres, 2021). A consequence of this problem is: even though the existence of both a “Tapejara-Sinopterus group” and of a “Thalassodromeus-Tupuxuara group” has been remarkably consensual, the same clades have received different names according to preferred phylogeny. Albeit valid under the ICZN, this situation is conflictive with the principles of phylogenetic nomenclature.

Under the light of phylogenetic nomenclature, it is undesirable that two equivalent clades should bear inconsistent names across distinct phylogenies. If distinct phylogenies agree on recovering a given clade (which is a great feat in pterosaur systematics), then this clade should have a consistent name, for the sake of stability. Different clade names should only exist when de facto distinct clade proposals exist. For example, a clade that includes Thalassodromeus and Azhdarcho but excludes Tapejara does not exist in certain propositions (e.g., Kellner, 2003). However, this clade exists in others (Unwin, 2003; Andres, 2021), under which such a proposed clade does need a name (“Neoazhdarchia”). Thus, Neoazhdarchia is a name that only exists (or is valid) within the context of a certain phylogenetic proposal (Unwin, 2003; Andres, 2021). In contrast, a clade that includes Sinopterus and Tapejara and excludes Thalassodromeus and Azhdarcho is universally accepted among pterosaur researchers. It is unfortunate that such welcome phylogenetic consensus is not accompanied by nomenclatural stability, as it should. It is for this reason that we adopt here the restrictive usage of Tapejaridae sensu Andres (2021), which has already been proposed and registered under the PhyloCode. This definition can be utilized in any phylogenetic proposal, and its adoption will prevent different workers from referring to different clades by, confoundingly, using the same names—as well as from referring to a same clade by different names.

Arguments for the restrictive usage of Tapejaridae sensu Andres (2021) need not come exclusively from the point of view of the PhyloCode, but could also be argued for under the ICZN. In the same way that the expansive Pteranodontidae sensu Bennett (1989, 1994) was elevated to the Pteranodontoidea of Kellner (2003), turning Pteranodontidae more restricted, then one might also regard that the original Tapejaridae sensu Kellner (1989, 2003) should be elevated to the Tapejaromorpha, with Tapejaridae becoming more restricted. We emphasize that the usage of these definitions as explored here do not imply, in any way, which phylogeny is preferred, and can stably be employed onto any presently existent phylogenetic proposal. In fact, the preferred proposal employed here is based on Pêgas et al. (2021), which is ultimately derived from Kellner (2003)—we corroborate the sister-group relationship between Tapejaridae and Thalassodromidae.

Results

Specimen-level variation survey

The generalized osteological pattern of Sinopterus complex specimens has already been described elsewhere (Zhang et al., 2019; Shen et al., 2021; Zhou, Niu & Yu, 2022). This section is not intended as a monographical account of the morphology of each specimen, but as a report of their most striking features, with particular focus on the anatomical variations we surveyed. Monographical descriptions are beyond the scope of the present paper and will be provided elsewhere. Specimens PMOL-AP00030 (Liu et al., 2015), SDUST-V1012 (Zhou, Niu & Yu, 2022) and SDUST-V1014 (Zhou et al., 2022) are not included in the present reassessment due to their rather incomplete nature. The holotype of Nemicolopterus crypticus, which may be a hatchling tapejarid (Witton, 2013; Naish, Witton & Martin-Silverstone, 2021), is also not included due to its very immature nature and disputed identification, and is thus discussed separately further below in the Discussion section.

Despite the relative completeness of several specimens, observation of anatomical details is rather limited due to preservational issues. As all specimens are crushed, bones are usually visible from a single side, sometimes obscured by overlaying bones, and sometimes too damaged, thus highly limiting comparisons. Osteological details are given below as possible. However, in most circumstances, details do not go further than gross shape seen from a single view (as demonstrated in our plates) and measurements. All specimens were measured first-hand, and raw measurements are presented in File S1 (Sheet 1). Specimens are presented below in chronological order of publication, from the oldest reported one to the most recently reported ones, and then finally with the ones reported here for the first time (D4019, BPMC 103, BPMC 104, BPMC 105, BPMC 106, and BPMC 107).

IVPP V 13363 (holotype of Sinopterus dongi)

Morphological survey

This specimen (Fig. 1) was originally described by Wang & Zhou (2003). It exhibits a relatively slender rostrum (~36% of jaw length), with a very low, incipient premaxillary crest and a low dentary crest. The rostrum is gently downturned at about 14° relative to the posterior occlusal line. The premaxillary crest is parabolical in outline. The nasoantorbital fenestra length/height ratio is not readily clear due to a slight anteroventral displacement of the orbitotemporal region. Still, it can be restored as somewhere between 2.8 and 3.2 (by restoring the position of the orbitotemporal region based on the inferred location of the quadratomandibular joint as indicated by the proportions of the mandible). The orbit has been described as subcircular (e.g., Andres, Clark & Xu, 2014), since its height and length are subequal. However, it may be described as subquadrangular due to the angular corners. This differs from the typical elongated piriform condition (higher than long, with a round dorsal margin and tapered ventral margin) of tapejarids and azhdarchoids in general (e.g., Kellner & Campos, 2007). Still, a tapered shape of the lower orbital margin is still present (in the jugal). The lacrimal process of the jugal is subvertical (only slightly anterodorsally oriented). A pair of slender, anteroventrally directed, and medially placed descending nasal processes is present. The posterior cranial crest processes (the posterior process of the premaxillae, and the frontoparietal crests) curve upwards. The quadrate is posteriorly reclined at ~160° relative to the palatal plane. The observable cervical formula is III < IV > V > VI > VII. The scapula is about 1.30 the length of the coracoid. The coracoid exhibits a clear ventral flange. The humeral deltopectoral crest is tongue-like and its long axis is sub-perpendicular relative to the long axis of the humeral shaft. The pteroid accounts for 43% of ulnar length. Metacarpal I is elongate, reaching the carpal region, while metacarpals II and III are reduced and restricted distally. Metatarsal I is the longest of the metatarsals (Wang & Zhou, 2003; Zhang et al., 2019).

Figure 1 Sinopterus dongi holotype (IVPP V 13363).

(A) Skeleton overview; (B) left metacarpus; (C) left foot; (D) skull (right lateral view). (E–H) Respective schematic drawings. Abbreviations: ca, carpus; co, coracoid; cv, cervical vertebra; d, dentary; d1–d4, digits 1–4; epi, epiphysis; etp, extensor tendon process; f, frontal; fe, femur; fpc, frontoparietal crest; h, humerus; ios, interorbital septum; l, left; lpt, lateral proximal tarsal; m, maxilla; mc, metacarpal; mt, metatarsal; n, nasal; naof, nasoantorbital fenestra; pm, premaxilla; ph, phalanx; pt, pteroid; ti, tibia; ul, ulna; r, right; rad, radius; sca, scapula; st, sternum. Scale bars: A, 50 mm; E, 50 mm; F, 20 mm; G, 10 mm; H, 20 mm.

Ontogenetic assessment

This specimen has already been regarded as a juvenile before (Kellner, 2010; Zhang et al., 2019). A large number of skeletal elements remain unfused in this specimen: scapulacoracoid, humeral epiphysis, carpal series, extensor tendon process of the first wing phalanx, and tibiotarsus. Several skull elements also remain unfused. It is clear that this specimen is a juvenile indeed, if compared to more ontogenetically advanced specimens in which the abovementioned elements are fused, such as in the postcranial skeleton of D2525 (File S1, Sheet 6). At a wingspan of 1.2 m, it would be conceivable that it was an advanced juvenile, older than smaller specimens such as the holotypes of S. gui (0.8 m), S. lingyuanensis (~0.85 m) and H. atavismus (0.85 mm, see further below), and younger than larger specimens such as the holotype of S. jii (1.6 m) and D2525 (2 m).

Remarks

This specimen is the holotype of Sinopterus dongi—the first genus and species of tapejarid to be described for the Jiufotang Fm. and Jehol Group as a whole. The validity of this genus and species has never been questioned.

BPV-077 (holotype of Sinopterus gui)

Morphological survey

The specimen (Fig. 2) is unfortunately badly preserved, with quite damaged and crushed bone surfaces (Li, Lü & Zhang, 2003). Still, general outlines of some of the skull and appendicular bones can be discerned. The skull is exposed mostly in left lateral view, except for the posterior region which seems to be broken and exposed in a slightly dorsolateral view. The rostrum accounts for ~39% of total jaw length. It is very slender (RI = 0.33) and crestless, while the dentary symphysis bears a very shallow crest. The nasoantorbital fenestra is very elongate (length/height ratio ~3.2). Quadrate inclination is unclear due to the bad preservation of the posterior region of the skull. Details of the cervical series are unclear due to bad preservation. The coracoid ventral margin bears a flange, similar to other Sinopterus complex specimens (see below). The deltopectoral crest of the humerus is rectangular, proximally placed, and bears a long axis roughly perpendicular relative to the main humeral shaft. The relative length of metacarpals I–III cannot be assessed. Of the wing fingers, only a first phalanx is preserved, thus obscuring wing phalanges proportions. Unfortunately, not much further details can be assessed due to the very limited preservational quality of the specimen.

Figure 2 Sinopterus gui holotype (BPV-077).

(A) Skeleton overview; (B) skull (left lateral view). (C and D) Respective schematic drawings. Abbreviations: co, coracoid; cv, cervical vertebra; d, dentary; dvs, dorsal vertebral series; f, frontal; fe, femur; fi, fibula; fpc, frontoparietal crest; h, humerus; is, ischium; j, jugal; l, left; mc, metacarpal; mt, metatarsal; n, nasal; naof, nasoantorbital fenestra; or, orbit; pt, pteroid; pu, pubis; prap, preacetabular process; ti, tibia; ul, ulna; r, right; rad, radius; sca, scapula. Scale bars: C, 50 mm; D, 50 mm.

Ontogenetic assessment

Unfused elements: palatal and posterior skull bones, dorsal centra and neural arches, scapula and coracoid, pelvic elements, tibia and fibula (entirely unfused). Other ontogenetic correlates cannot be assessed. This specimen is clearly a very young juvenile. It is also the second smallest of all Jehol tapejarid specimens (second to the holotype of Nemicolopterus crypticus), with an estimated wingspan of only 64 cm (Kellner & Campos, 2007).

Remarks

This specimen is the holotype of Sinopterus gui—the second species of tapejarid to be described for the Jiufotang Fm. and Jehol Group as a whole (Li, Lü & Zhang, 2003). It was subsequently recognized as a very young juvenile (Kellner & Campos, 2007). The validity of this species has been questioned several times, in all such cases being regarded as a junior synonym of S. dongi even when multiple Jiufotang tapejarid species were accepted, on the basis that it could not be distinguished from S. dongi (Kellner & Campos, 2007; Kellner, 2010; Zhang et al., 2019). This is problematic because recent publications have simply repeated the interpretation of S. gui being indistinguishable from S. dongi while not comparing S. gui to other more recently named species considered as valid, thus not justifying why it is indistinguishable from S. dongi only and not from any further species (e.g., Zhang et al., 2019). First described by Li, Lü & Zhang (2003), these authors recognized it as distinct from Sinopterus dongi at a species-level, yet sufficiently similar to be placed in the same genus. Originally, Li, Lü & Zhang (2003) proposed the following diagnosis for the new species: “[e]leven dorsal vertebrae fused into notarium, and they are nearly equal in length. At least four sacral vertebrae, humerus longer than scapula, wing metacarpal slightly shorter than the first wing phalange, the distal end of the deltopectoral process not expanded, ratio of the femur to the tibia is approximately 0.49” (Li, Lü & Zhang, 2003: p. 445). Later, Kellner & Campos (2007) observed that this specimen does not present a notarium (which is an advanced ontogenetic feature). Instead, it represents a very young, juvenile specimen (Kellner & Campos, 2007; Kellner, 2010). Most authors have, since then, been unable to distinguish S. gui from S. dongi, and thus interpreted the holotype of Sinopterus gui as a juvenile specimen of Sinopterus dongi (e.g., Kellner & Campos, 2007; Zhang et al., 2019), although Kellner (2010) noticed that it could represent a juvenile of some other Jiufotang tapejarid instead, such as Huaxiapterus corollatus (therein referred to as Sinopterus corollatus). The interpretation of the holotype of S. gui as a juvenile of S. dongi (and not any other Jiufotang tapejarid species) has been maintained by Zhang et al. (2019) without further justifications, even though these authors accept the validity of several other Sinopterus species (S. lingyuanensis, S. corollatus, S. benxiensis, and S. atavismus). We maintain here that S. gui is indeed indistinguishable from S. dongi except for the complete absence of a premaxillary crest in the former, which is easily attributed to ontogeny (Witton, 2013; Zhang et al., 2019).

GMN-03-11-001 (holotype of Huaxiapterus jii)

Morphological survey

This almost complete specimen includes a partial skull, although the posterior region is disarticulated and damaged (Fig. 3). The rostrum is ventrally deflected at 14° relative to the posterior palatal plane. The rostrum exhibits a premaxillary crest. It is similar in shape to that of S. dongi (parabolical in outline), despite being larger. It is distinct from the premaxillary crest condition of other proposed species, such as the pointed premaxillary crests of Huaxiapterus atavismus (both specimens, the holotype XHPM 1009 and the referred specimen IVPP V 22338) or the trapezoidal crests of Huaxiapterus corollatus and Huaxiapterus benxiensis, or the crestless conditions seen in Sinopterus gui and Sinopterus lingyuanensis. Most of the posterior region of the skull is badly damaged, except for the left jugal which is partially preserved. The jugal is triradiate, unlike the tetraradiate condition seen in Tapejara wellnhoferi (Wellnhofer & Kellner, 1991), Caiuajara dobruskii (Manzig et al., 2014) and Tupandactylus navigans (Beccari et al., 2021). The lacrimal and postorbital processes of the jugal describe a roughly perpendicular angle. The proportions of the nasoantorbital fenestra cannot be readily measured due to the damaged nature of the posterior region of the skull, but an estimate can still be given based on the location of the lacrimal process of the jugal (about three times as long as high). As with the premaxillary crest, the dentary crest is also larger than in S. dongi. Only two disarticulated cervical vertebrae can be seen, so that the cervical formula cannot be assessed. Pteroid length is equivalent to about 44% of the ulna length. Metacarpal I is elongate, extending for at least 90% the length of metacarpal IV. Wing proportions are closest to the holotype of S. dongi (Fig. 3; File S1, Sheets 1, 3). Pedal elements are entirely disarticulated, so that the metatarsal formula cannot be assessed.

Figure 3 Huaxiapterus jii holotype (GMN-03-11-001).

(A) Skeleton overview; (B) skull (left lateral view, slightly ventrolateral). (C and D) Respective schematic drawings. Abbreviations: ca, carpus; co, coracoid; cv, cervical vertebra; d, dentary; d1–d4, digits 1–4; dv, dorsal vertebra; epi, epiphysis; fe, femur; h, humerus; j, jugal; l, left; mc, metacarpal; pm, premaxilla; ph, phalanx; pp, prepubis; pt, pteroid; ti, tibia; ul, ulna; r, right; rad, radius; sca, scapula; st, sternum. Scale bars: C, 50 mm; D, 20 mm.

Ontogenetic assessment

The holotype of S. jii has been regarded as a juvenile compatible with the holotype of S. dongi, given their similarity in lacking bone fusion between posterior skull elements, scapulocoracoid, humeral epiphyses, carpals, extensor tendon process of the first wing phalanx, and tibiotarsus (Kellner, 2010). However, it is worth noticing that the dorsal centra and arches of GMN-03-11-001 are partially fused (they bear a visible suture, but are not found disassociated), unlike some entirely unfused and disassociated dorsal centra and arches seen in the holotype of S. dongi. This suggests that GMN-03-11-001 is slightly more ontogenetically developed than the holotype of S. dongi, both as juveniles. Concerning body size, GMN-03-11-001 is larger than the holotype of S. dongi, with a humerus of 79 mm in length and a wingspan of 1,602 mm.

Remarks

This specimen was originally described as representing a new genus and species, Huaxiapterus jii (Lü & Yuan, 2005). Subsequent publications have considered it either as a species of Sinopterus, as S. jii (Kellner & Campos, 2007; Pinheiro et al., 2011; Kellner, 2013), or as a junior synonym of Sinopterus dongi (Wang & Zhou, 2006; Witton, 2013; Zhang et al., 2019), thus invalidating the genus Huaxiapterus. Still, other researchers still considered H. jii as valid and as a distinct taxon, with the genus Huaxiapterus being valid (Andres, Clark & Xu, 2014; Lü et al., 2016).

This taxon was originally diagnosed based on cranial crest development: premaxillary and dentary crests deeper than in Sinopterus dongi and shallower than in Tapejara wellnhoferi (see Lü & Yuan, 2005), though without precise quantitative comparisons. Later, this species has been regarded as a junior synonym of Sinopterus dongi: Wang & Zhou (2006) were unable to find differences between the holotypes of the two species, and thus synonymized them. At the time, these two species (together with Sinopterus gui) were the only named species within the Sinopterus complex. We maintain that the holotypes of S. gui and S. jii are indistinguishable from S. dongi, and further add that S. jii shares with S. dongi the following features: metacarpal I articulating with the carpus, and wing phalanx 4/phalanx 1 length ratio about ~0.30, which distinguish these proposed taxa from other proposed taxa such as H. corollatus and H. benxiensis (see below). Sadly, these features are uncertain in the holotype of S. gui.

ZMNH M813 (holotype of Huaxiapterus corollatus)

Morphological survey

This specimen is almost complete, although some skeletal regions are badly damaged and anatomical details are obliterated, particularly the posterior region of the skull, post-cervical vertebrae, and the pedes (Fig. 4). The skull exhibits a trapezoidal premaxillary crest and a shallow dentary crest. The rostrum is relatively robust, akin to that of the holotype of Huaxiapterus jii and unlike the holotypes of S. dongi or S. gui. The rostrum is ventrally deflected by 21° (contra 14° in the holotypes of S. dongi and H. jii). The nasoantorbital fenestra is relatively short, with an estimated length/height ratio of about 2.2 (based on its length as inferred from the location of the quadratomandibular joint, as indicated by the preserved mandible, as it roughly correlated to the posterior margin of the nasoantorbital fenestra in sinopterines and tapejarids overall; e.g., Kellner & Campos, 2007; Kellner, 2013; Lü et al., 2016). A clear occlusal gap is present between the dentary and the rostrum (as originally indicated, see Lü et al., 2006a), unlike what has been represented in some reconstructions (e.g., Witton, 2013). The cervical series is partially obscured by the radius and ulna, which lay over cervicals IV–V, hindering assessment of their relative lengths. Metacarpals I–III are reduced, and it can be seen that metacarpals I and II do not contact the carpus, reaching only about a third of the length of metacarpal IV. Wing proportions deviate from previously reported specimens in that the fourth wing phalanx is relatively shorter, accounting for only ~20% of the first phalanx (contra ~30% in the holotypes of S. dongi and S. jii).

Figure 4 Huaxiapterus corollatus holotype (ZMNH M813).

(A) Skeleton overview; (B) skull (right lateral view); (C) left metacarpus. (D–F) Respective schematic drawings. Abbreviations: ca, carpus; co, coracoid; cv, cervical vertebra; d, dentary; d1–d4, digits 1–4; fe, femur; fpc, frontoparietal crest; h, humerus; l, left; m, maxilla; mand, mandible; mc, metacarpal; mt, metatarsal; n, nasal; naof, nasoantorbital fenestra; pm, premaxilla; ph, phalanx; pt, pteroid; ti, tibia; ul, ulna; r, right; rad, radius; sca, scapula; sk, skull. Scale bars: A, D, 100 mm; E, F, 10 mm.

Ontogenetic assessment

In this specimen, unfused skeletal elements include the posterior skull bones, scapulocoracoid, and extensor tendon process of the first wing phalanx. Unfortunately, fusion of humeral epiphyses cannot be assessed due to poor preservation. The tarsals are fused to the tibia, forming a tibiotarsus, as can be seen from the right hindlimb. The carpals also seem to be fused into distal and proximal syncarpals. Thus, this specimen seems to be relatively more mature than the holotypes of S. dongi, S. gui and H. jii, as a subadult. It is roughly equivalent to the holotype of H. jii, with a humerus of 75 mm in length and a wingspan of 1,560 mm.

Remarks

This specimen was designated as the holotype of Huaxiapterus corollatus by Lü et al. (2006a). The species-level validity of this species (irrespective of its generic status) has been mostly accepted (Pêgas, Leal & Kellner, 2016; Lü et al., 2016; Zhang et al., 2019; Andres, 2021), except for Witton (2013) who preliminarily proposed that all Jiufotang tapejarids were synonymous with S. dongi. It is interesting to note that, although Naish, Witton & Martin-Silverstone (2021) preliminarily corroborated Witton (2013) view, they highlighted that at least the holotype of H. corollatus could potentially represent a new taxon (based on its limb proportions), pending further study.

The taxon Huaxiapterus corollatus was originally diagnosed on the basis of cranial crest features, namely crest shape (“hatchet-shaped”), position (level with the anterior margin of the nasoantorbital fenestra), and orientation (“short axis perpendicular to the anterodorsal margin of the nasoantorbital fenestra”; see Lü et al., 2006b). These conditions differ starkly from what is seen in the holotypes of S. dongi, S. gui and H. jii. However, as noticed by Witton (2013) and Naish, Witton & Martin-Silverstone (2021), cranial crest features used alone make for dangerous taxonomic decisions, as they could rather reflect ontogenetic or sexual variations. Still, the holotype of H. corollatus also differs from the holotypes of S. dongi and S. jii in exhibiting a reduced metacarpal I, and in wing proportions (File S1, Sheet 1). H. corollatus exhibits a reduced wing phalanx 4, which accounts for ~20% of the length of the first wing phalanx, contra ~30% in the previously named S. dongi and H. jii. Naish, Witton & Martin-Silverstone (2021) noticed that the holotype of H. corollatus was an apparent outlier within the Sinopterus complex regarding limb proportions, leading them to propose that it could be a potentially valid taxon pending further study.

D2525

Morphological survey

D2525 is an almost complete postcranial skeleton, lacking the skull, part of the anterior cervical series, part of the posterior dorsal series, and the sacral and caudal series (Fig. 5). The preserved cervical vertebrae, as well as shoulder girdle and right humerus, are badly damaged. Although previously unreported, the ?fourth cervical (exposed in ventral view, retaining some tridimensionality) clearly exhibits a pneumatic foramen piercing its lateral surface. The sternum is approximately square, with the posterior margin convex. The left coracoid bears a well-developed ventral flange. The left humerus is exposed in dorsal view, and no dorsal proximal pneumatic foramen can be seen in this specimen, as in IVPP V 23388 (Zhang et al., 2019). The ulnar crest is rounded. The humeral shaft is mostly straight, except for the distal portion which is slightly anteriorly recurved. Metacarpals I–III are tightly appressed to metacarpal IV on the distal metacarpal region on both sides. Metacarpal I extends for only about 40% of the length of metacarpal IV (Fig. 5). Wing proportions are very similar to the holotypes of H. corollatus and H. benxiensis, with the fourth wing phalanx corresponding to ~20% the length of the first wing phalanx (contra ~30% in S. dongi and S. jii). Wing phalanges are exposed in ventral view, and a longitudinal ridge can be seen in phalanges 2 and 3, similarly to H. atavismus (Lü et al., 2016) and IVPP V 23388 (Zhang et al., 2019). In the pedes, metatarsal I is distinctively shorter than metatarsal II, which is the longest.

Figure 5 Specimen D2525.

(A) Skeleton overview; (B) right foot; (C) right metacarpus. (D–F) Respective schematic drawings. Abbreviations: co, coracoid; cv, cervical vertebra; d, dentary; d1–d4, digits 1–4; dsc, distal synpcarpal; etp, extensor tendon process; f, frontal; fe, femur; fpc, frontoparietal crest; gas, gastralia; h, humerus; ios, interorbital septum; is, ischium; l, left; lpt, lateral proximal tarsal; mc, metacarpal; mt, metatarsal; pc, preaxial carpal; ph, phalanx; poap, postacetabular process; pp, prepubis; prap, preacetabular process; psc, proximal syncarpal; pt, pteroid; pu, pubis; ti, tibia; ul, ulna; r, right; rad, radius; ri, rib; sca, scapula; st, sternum. Scale bars: D, 50 mm; E, 10 mm; F, 50 mm.

Ontogenetic assessment

Specimen D2525 is the third largest of all known Jiufotang tapejarids, with a 2-m wingspan (Lü et al., 2006b), and also appears to be one of the most osteologically mature ones. Observable fused elements include dorsal neural arches and centra, the scapulocoracoid, the syncarpals, and the extensor tendon process of the first wing phalanx. Partial fusion (almost complete fusion, with faint indications of sutures) can also be seen in the pelvis, tibiotarsus, and tarsal elements. The presence of a notarium cannot be assessed due to preservational limitations, since the anterior dorsal series is preserved in ventral view and badly crushed.

Remarks

This specimen was originally described as a new specimen of Sinopterus dongi, based on the assertion that the limb proportions of D2525 were most similar to S. dongi than to S. gui, H. jii or H. corollatus, which were the four existing nominal species at the time (Lü et al., 2006b). Such referral has never been contested in the literature. Contrary to previous reports (Lü et al., 2006b), the limb proportions of D2525 are most similar to the holotype of H. corollatus, and not S. dongi (see File S1, Sheets 1, 3). In fact, D2525 is herein considered as indistinguishable from H. corollatus, with which it shares a shortened metacarpal I (about 40% the length of metacarpal IV, contra >90% in S. dongi and S. jii) and a shortened fourth wing phalanx (~20% of first phalanx length, contra ~30% in S. dongi and S. jii). It differs from the holotypes of S. dongi and H. jii in wing proportions and in metatarsals I–II relative length (metatarsal II is the longest one in D2525, instead of metatarsal I as in S. dongi).

BXGM V0011 (holotype of Huaxiapterus benxiensis)

Morphological survey

This specimen consists on a virtually complete specimen (Fig. 6). However, some anatomical regions are damaged and/or partially obscured, mainly the torso region (with the post-cervical vertebral series, sternum, ribs, and scapulocoracoid). The rostrum is built similarly to the holotype of H. corollatus, with a downward deflection of 20°. The premaxillary crest is slightly larger than in the holotype of H. corollatus, but it is similar in being distinctively anterodorsally protrusive with abrupt limits, unlike the smoothly-transitioning borders of the parabolical crests of the holotypes of S. dongi and S. jii. Despite broken, the premaxillary crests seems to have been trapezoidal in shape, as in the holotype of H. corollatus. The posterior process of the premaxillae is steeply dorsally recurved. An elongate posterior spine (posterior process of the premaxillae + frontoparietal crest) is present, much larger than in the holotype of S. dongi. The nasoantorbital fenestra is approximately as elongate as in S. dongi, with a length/height ratio of about 2.4. The long axis of the nasal process is very deflected anteriorly, unlike the almost verticalized nasal process seen in the holotype of S. dongi. The shape of the jugal (as seen from the lacrimal and postorbital processes) demonstrates that the orbit was piriform, with a tapered ventral margin, and quite higher than wide, unlike the subquadrangular orbit of S. dongi. The quadrate is posteriorly inclined at about 153°. Not much further detail can be seen due to extensive superficial damage. The observable cervical formula is III < IV < V > VI. Both humeri are badly damaged, with only a section being exposed. The original description reported on an oddly short humerus only 55% the length of the femur (Lü et al., 2007), but this seems to have been based on the fairly incomplete right humerus. We reidentify here the damaged proximal and distal limits of the left humerus, which indicate it was comparable to that of other Jiufotang tapejarids (about 80% of femur length) instead of oddly short (Figs. 6A and 6D). The extension of the pteroid is unclear. Metacarpal I confidently extends for only ~40% the length of metacarpal IV. The proximal extension of metacarpals II and III is unfortunately obscure, since it is unclear if the proximal tips are broken or not. Wing proportions closely match H. corollatus, with relatively short fourth wing phalanges (20% the length of the first phalanx). The relative length of metatarsals I–III overall cannot be assessed due to poor preservation.

Figure 6 Huaxiapterus benxiensis holotype (BXGM V0011).

(A) Skeleton overview; (B) left metacarpus; (C) skull (left lateral view). (D–F) Respective schematic drawings. Abbreviations: cv, cervical vertebra; d, dentary; d1–d4, digits 1–4; f, frontal; fe, femur; fpc, frontoparietal crest; h, humerus; ios, interorbital septum; j, jugal; l, left; m, maxilla; la, lacrimal; mc, metacarpal; mt, metatarsal; naof, nasoantorbital fenestra; np, nasal process; pm, premaxilla; ph, phalanx; pt, pteroid; ti, tibia; ul, ulna; r, right; rad, radius. Scale bars: 50 mm.

Ontogenetic assessment

In this specimen, fused skeletal elements include the humeral epiphyses, syncarpals, the extensor tendon process of the first wing phalanx, and the tibiotarsus. Scapulacoracoids cannot be observed due to damage. Only the posterior skull bones are still unfused to the rest of the skull. Indeed, posterior skull bones are known to be among the last skeletal elements to fuse in pterosaurs (e.g., Kellner, 2015). This specimen thus exhibits a relatively advanced level of skeletal fusion, fitting well with the concept of an advanced subadult among pterosaurs (e.g., Kellner & Tomida, 2000). This specimen is clearly one of the most mature ones in the present sample, along with D2525 (see above), since all of the previously described specimens lack fusion of the extensor tendon process of the first wing phalanx. It is roughly equivalent in body size to the holotypes of H. jii and H. corollatus, with a humerus of 82 mm in length and a wingspan of 1,600 mm.

Remarks

The species H. benxiensis was erected on the basis of BXGM V0011 and attributed to the genus Huaxiapterus, following H. jii and H. corollatus. The validity of this species has been mostly accepted without further comments (Pinheiro et al., 2011; Kellner, 2013; Pêgas, Leal & Kellner, 2016; Zhang et al., 2019; Andres, 2021), except for works that argued for the “restrictive taxonomic scheme” of the Sinopterus complex, which regarded it as most likely a junior synonym of S. dongi along with all other nominal species of Jiufotang tapejarids (Witton, 2013; Naish, Witton & Martin-Silverstone, 2021).

Huaxiapterus benxiensis has been regarded as distinct from H. corollatus on the basis of an “elongate parietal spine”, “well-developed premaxillary crest”, and a shallow groove on the occlusal surface of the dentary symphysis (Lü et al., 2007). Witton (2013) noticed that crest-related features could be influenced by ontogeny rather than interspecific variation. We further note that the “shallow groove” on the anterior end of the symphysis corresponds to the anterior occlusal depression (ubiquitous to tapejarids), interrupted posteriorly by a transverse ridge (similar to the condition seen in Bakonydraco galaczi; see Ősi, Weishampel & Jianu, 2005). This condition can also not set H. benxiensis apart from any other proposed Jehol tapejarid species, since preservation precludes the verification of this feature in other type specimens. H. benxiensis is here considered as indistinguishable from H. corollatus, with which it shares a rostrum deflection of ~20°, a reduced metacarpal I, and a reduced fourth wing phalanx (~20% of first wing phalanx length). Both H. benxiensis and H. corollatus further differ from S. dongi and S. gui in exhibiting a relatively shorter nasoantorbital fenestra (only 2.2–2.4 in height/length ration, contra ~3 in S. dongi and S. gui).

JPM-2014-005 (holotype of Sinopterus lingyuanensis)

Morphological survey

The holotype of S. lingyuanensis exhibits a relatively fine preservation, comprising an almost complete skeleton lacking only some distal wing phalanges and the tail. Some anterior trunk and appendicular elements, such as posterior cervical vertebrae, some dorsal vertebrae, ribs, sternum, and pectoral girdle, are severely crushed against each other and cannot be discerned (Fig. 7). Other than that, most other skeletal elements are discernible, with decent surface preservation despite crushing. The skull is exposed mainly in left lateral view, and the occipital region is laterally displaced towards the left, thus being visible in a somewhat posterolateral view. The rostrum is entirely crestless and slender, accounting for 44% of total jaw length. The rostrum is gently deflected at 12° relative to the palatal plane. Beneath the anterior level of the nasoantorbital fenestra, a bulge is present on the jaw margin, indicating the presence of a slight lateral palatal expansion similar to what is seen in Tapejara and Caiuajara (Wellnhofer & Kellner, 1991; Manzig et al., 2014). The nasoantorbital fenestra is quite elongate, being 3.25 times longer than high. The nasals exhibit a pair of descending nasal processes, which are subvertical and elongate, similar to S. dongi and unlike the anteriorly directed, short condition seen in H. benxiensis. The orbit is roughly subquadrangular, about as wide as high, similarly to S. dongi. The divergence angle between the lacrimal and postorbital processes of the jugal is about ~90°, similar to S. dongi and H. jii but unlike H. benxiensis (~68°), which exhibits a piriform orbit. The quadrate is reclined at about 160°. A small, short frontoparietal crest is present, extending beyond the occiput. The mandible is exposed in dorsal view. Sadly, the occlusal surface is not well-preserved. Still, it can be seen that a slight lateral expansion occurs at the posterior region of the symphysis, as in Tapejara and Caiuajara (Wellnhofer & Kellner, 1991; Manzig et al., 2014), matching the slight lateral palatal expansion beneath the anterior margin of the nasoantorbital fenestra. The dentary symphysis and the retroarticular process account for, respectively, 53% and 4% of total mandibular length. Atlas and axis cannot be observed. The observable cervical formula is III < IV > V > VI, similar to S. dongi and unlike H. benxiensis in which the fifth cervical is the longest. The mid-cervicals clearly exhibit at least one pneumatic foramen piercing their lateral sides. The pteroid accounts for 47% of ulnar length. Sadly, the distal extensions of metacarpals I–III are obscured by metacarpal IV. The relative length of the fourth wing phalanx is also unknown. In the pedes, the metatarsal formula is I < II > III > IV, similar to D2525 but unlike S. dongi.

Figure 7 Sinopterus lingyuanensis holotype (JPM-2014-005).

(A) Skeleton overview; (B) right foot; (C) skull (left lateral view). (D–F) Respective schematic drawings. Abbreviations: art, articular; ca, carpus; cv, cervical vertebra; d, dentary; d1–d4, digits 1–4; fe, femur; fpc, frontoparietal crest; h, humerus; ios, interorbital septum; j, jugal; l, left; lpt, lateral proximal tarsal; mc, metacarpal; mt, metatarsal; naof, nasoantorbital fenestra; np, nasal process; pm, premaxilla; ph, phalanx; pt, pteroid; pv, pelvis; t, tarsus; ti, tibia; ul, ulna; r, right; rad, radius; sca, scapula; sv, sacral vertebrae. Scale bars: D, 50 mm; E, 10 mm; F, 50 mm.

Ontogenetic assessment

JPM-2014-005 is small-sized, with a skull length of 112 mm and an estimated totalized wingspan of ~850 mm. Skull elements are mostly unfused, to the exception of the premaxillomaxillae and dentaries. Postcranial unfused elements include the humeral epiphyses, carpals, extensor tendon process of the first wing phalanx, pelvic elements, and tibiotarsus. Fusion (or lack thereof) of further elements cannot be assessed. The available information suggests JPM-2014-005 is a young juvenile, as the holotypes of S. gui and S. dongi.

Remarks

This specimen was originally designated as the holotype of a new species, S. lingyuanensis, by Lü et al. (2016). This was subsequent to Witton (2013) proposition that all Jiufotang tapejarids formed an ontogenetic continuum of S. dongi, which was not accepted by Lü et al. (2016). Later, Zhang et al. (2019) expressed their approval over the validity of S. lingyuanensis, without further comments. Later, Naish, Witton & Martin-Silverstone (2021) echoed the proposition of Witton (2013) that all proposed Jiufotang tapejarids most likely represented a single species (to the potential exclusion of H. corollatus), including S. lingyuanensis.

The species Sinopterus lingyuanensis was proposed based on the following features: nasoantorbital fenestra length/height ratio 3.2, rostral index 3.03, femur/tibia length ratio 0.66, and wing phalanx 2/wing phalanx 1 length ratio 0.85 (Lü et al., 2016). However, all of these values fit well within the spectrum seen in the Sinopterus complex (File S1, Sheet 1) and cannot set S. lingyuanensis apart from other species, particularly from S. dongi, S. gui and H. jii which also exhibit nasoantorbital fenestra about three times as long as high (distinct in this regard from the holotypes of H. corollatus and H. benxiensis). Still, S. lingyuanensis does differ from S. dongi in metatarsal configuration (I ≈ II, rather than I > II), and also differs from H. benxiensis in orbit shape (subcircular rather than piriform), nasal descending process configuration (subvertical and elongate, rather than anteriorly directed and short), and cervical formula (IV > V, rather than IV < V). It also differs from both H. corollatus and H. benxiensis in exhibiting a gentler rostrum deflection (12° rather than 20°). The significance of these variations will be discussed further below, in the Discussion section.

XHPM 1009 (holotype of Huaxiapterus atavismus)

Morphological survey

Despite virtually complete, many skeletal remains of this specimen are quite jumbled together, preventing the observation of much anatomical data (Fig. 8). The rostrum exhibits a very small, triangular-shaped premaxillary crest, whose apex is anterodorsally oriented and located posterior to the anterior margin of the nasoantorbital fenestra (this configuration is distinct from any other tapejarid specimen previously published, but similar to specimens IVPP V 23388 and D4019). The rostrum is slender, ventrally deflected by 14°, and with a deflection point anteriorly located, similarly to S. lingyuanensis. A small, yet clearly perceivable, occlusal gap is present. The dentary bears a slight dorsal eminence, as well as a low dentary crest. The observable cervical formula is III < IV > V ≅ VI > VII > VIII. Not much can be discerned from the remaining of the axial skeleton, and the same is true for the pectoral girdle. The pteroid accounts for 40% of ulnar length. Unfortunately, the relative lengths of the metacarpals cannot be assessed. Wing phalanx proportions are a close match for S. dongi and S. jii (File S1, Sheet 1), and distinct from H. corollatus, H. benxiensis and D2525 which exhibit a comparatively reduced fourth wing phalanx about 20% the length of the first wing phalanx (File S1, Sheet 1). Metatarsal I is shorter than metatarsal II, which is the longest, unlike S. dongi.

Figure 8 Huaxiapterus atavismus holotype (XHPM 1009).

(A) Skeleton overview; (B) skull (left lateral view). (C and D) Respective schematic drawings. Abbreviations: ca, carpus; cv, cervical vertebra; co, coracoid; d, dentary; d1–d4, digits 1–4; fe, femur; fpc, h, humerus; mand, mandible; mc, metacarpal; mt, metatarsal; pmc, premaxillary crest; ph, phalanx; ti, tibia; ul, ulna; r, right; rad, radius; sk, skull. Scale bars: 50 mm.

Ontogenetic assessment

XHPM 1009 is a small-sized specimen, with an estimated skull length of ~120 mm and total wingspan of ~850 mm. Unfused skeletal elements include the carpals, extensor tendon process of the first wing phalanx, and tibiotarsus. Sadly, not much else can be discerned. Still, this specimen is compatible with a young juvenile, not much more advanced than the holotype of S. gui.

Remarks

This specimen was originally designated as the holotype of a new species, H. atavismus, by Lü et al. (2016). This was subsequent to Witton (2013) proposition that all Jiufotang tapejarids formed an ontogenetic continuum of S. dongi. Still, Zhang et al. (2019) accepted the validity of this species, which they assigned to the genus Sinopterus, as Sinopterus atavismus. Later, Naish, Witton & Martin-Silverstone (2021) echoed the proposition of Witton (2013) in interpreting all Jiufotang tapejarids as probable synonyms, to the inclusion of S. atavismus.

The species H. atavismus was originally diagnosed based on the presence of a squared premaxillary crest and of a ventral groove on the second wing phalanx. As noticed by Zhang et al. (2019), the crest is actually not squared (Fig. 8), and cranial crest morphology should be viewed with caution when discussing pterosaur diagnoses; while the ventral groove on the second wing phalanx is probably common within tapejarids (see Kellner, 2004; Zhang et al., 2019), although admittedly hard to ascertain in other Sinopterus complex specimens due to heavy crushing. H. atavismus shares with S. dongi and S. lingyuanensis a fourth cervical vertebra longer than the fifth, distinct from H. benxiensis and other tapejarids. H. atavismus differs from the holotype of S. dongi in pedal morphology, showing the typical condition (metatarsal II the longest), and not the unique condition seen in S. dongi (metatarsal I the longest). H. atavismus differs from H. corollatus and H. benxiensis in exhibiting a gentler rostrum deflection and a more elongate fourth wing phalanx (File S1, Sheet 1), and from D2525 in the latter aspect as well.

IVPP V 23388

Morphological survey

This specimen has been described and figured in detail by Zhang et al. (2019). The rostrum is elongate and slender, with a gentle ventral deflection of 14°. The rostrum deflection point lies anterior to the anterior margin of the nasoantorbital fenestra, as in S. lingyuanensis and H. atavismus. The premaxilla produces a small, subtriangular crest, as noted by Zhang et al. (2019), similar to that seen in the holotype of H. atavismus. Despite the incomplete, disarticulated nature of the skull remains, the nasoantorbital fenestra is notoriously elongate, and was confidently over three times as elongate as high (Zhang et al., 2019). The jugal is triradiate, and the angle formed between the lacrimal and postorbital processes is very wide (~90°, similar to S. dongi and S. lingyuanensis), indicating the orbit was probably subquadrangular in shape, and not ventrally tapered (piriform) as in H. benxiensis. The postoccipital extension of the premaxillae is elongate and curved posterodorsally. The observable cervical formula is IV > V ≅ VI > VII > VIII > IX (contra Zhang et al., 2019). The coracoid exhibits a deep ventral flange proximally. Metacarpals II and III are reduced, while the preserved metacarpal I extends for about 85% the length of metacarpal IV. The proximalmost tip of metacarpal I is missing due to a crack in the slab. Sadly, pteroid and wing phalanges 4 are missing. Metatarsal I is shorter than metatarsal II, which is the longest.

Ontogenetic assessment

With a humerus of 79 mm in length and a wingspan of ~1,600 mm, this specimen is similar in size to the holotype of H. benxiensis, and amongst the largest specimens in the Sinopterus complex. Partially fused elements include the carpals, pubis and ischium, ilium and pubosichiadic plate, notarium and synsacrum, and tibiotarsus—these elements are tightly bound, though with faint, visible sutures. Unfused elements include the scapulocoracoid, humeral epiphysis extensor tendon process, and orbitotemporal bones. Despite not an adult, this specimen is clearly more mature than the juvenile holotypes of S. gui, S. dongi, S. lingyuanensis and H. atavismus, and could be considered a subadult.

Remarks

This specimen has been attributed to Sinopterus atavismus (=Huaxiapterus atavismus) by Zhang et al. (2019). No alternative attributions have been given by any other workers, except for Naish, Witton & Martin-Silverstone (2021) who preliminarily considered that all Jiufotang tapejarids were most likely conspecific with S. dongi (to the potential exception of H. corollatus only).

This fairly complete specimen was described recently by Zhang et al. (2019), who were unable to distinguish it from Huaxiapterus atavismus and thus referred the new specimen to this species (using the combination Sinopterus atavismus). Zhang et al. (2019) considered that three features allowed IVPP V 23388 to be identified as H. atavismus: the shape of the premaxillary crest, the shape of the anterodorsal margin of the premaxilla, and the proportions between metatarsals I and II (Zhang et al., 2019). However, the first two features are influenced by the development of the premaxillary crest, which, as discussed above, is prone to sexual and ontogenetic variation, and should be viewed with caution before being utilized in diagnoses, as will be discussed further below in this work.

Furthermore, proportions between metatarsals I and II in IVPP V 23388 (metatarsals I/II = ~0.90) and the holotype of H. atavismus are rather close to those of other specimens such as S. lingyuanensis (File S1, Sheets 1, 3), and thus this condition should be seen with caution. These three specimens also match well in the configuration of the nasoantorbital fenestra (over three times as long and high) and rostrum deflection angle (12°–14°), also matching S. dongi and H. jii in these regards, being all distinct from H. corollatus and H. benxiensis (with nasoantorbital fenestrae about 2.3 times as long as high, and rostrum deflections of 20°–21°). We regard that IVPP V 23388, along with the holotype of H. atavismus, are both indistinguishable from S. lingyuanensis. They are all also undistinguishable from S. dongi except for the metatarsi proportions.

D3072

Morphological survey

This specimen has been recently described and figured in detail by Shen et al. (2021). It consists of a partial postcranial skeleton, comprising most of the cervical and dorsal series, the forelimbs, and partial hindlimbs. The observable cervical formula is III < IV > V > VI > VII > VIII > IX. Single pneumatic foramina can be seen piercing the lateral sides of some cervical vertebrae (at least III, IV and V; unclear in others). Metacarpal I is elongate, with a preserved portion accounting for about 90% of metacarpal IV length; the proximal tip is missing and it may have been longer. The first wing phalanx exhibits two pneumatic foramina piercing the ventral side of the proximal region, similar to Keresdrakon vilsoni (see Kellner et al., 2019). The fourth phalanx is relatively large, accounting for 36% the length of the first wing phalanx, approaching more closely the value seen in the holotype of S. dongi and in IVPP V 23388 (30%). In the pedes, metatarsal I is the longest one.

Ontogenetic assessment

As originally indicated by Shen et al. (2021), this specimen is clearly a juvenile as seen from the lack of fusion between many skeletal elements: the humeral epiphyses, scapulocoracoid, the extensor tendon process of the first wing phalanx, the carpal elements, tibia and fibula, tibia and proximal tarsals, and neural arches and centra of most dorsal vertebrae. Only the neural arches and centra of cervical vertebrae and anterior dorsal vertebrae are fused. With a humerus of 55 mm in length and wingspan of 1,135 mm, this specimen is similar in size to the holotype of S. dongi (humerus 58 mm in length and wingspan of 1,200 mm), which is also interpreted as a juvenile.

Remarks

This specimen has been referred to S. dongi by Shen et al. (2021), as accepted by Zhou et al. (2022) and not commented on the literature any further so far. Shen et al. (2021) noticed that D3072 shares with the holotype of S. dongi similar limb proportions as well as a reduced metatarsal I (shorter than metatarsals II and III), which has been considered a diagnostic apomorphy for S. dongi within the expansive taxonomic scheme of the Sinopterus complex (Zhang et al., 2019).

D4019 (new specimen)

Morphological survey

This specimen comprises an almost complete skeleton, although not very well preserved. Many of the elements are articulated, except for most skull and manual elements (Fig. 9). The rostrum is slender and gently decurved (by 13°) and bears a well-developed, heaped crest. The dorsal margin of the premaxilla is slightly jagged. The jugal-quadratojugal-quadrate complex indicates the quadrate was strongly reclined (by 162°). Unfortunately, the jugal is incompletely preserved and lacks a lacrimal process. A well-developed and posterodorsally inclined frontoparietal crest is present. The cervical vertebrae not very well-preserved and not much can be observed beyond their lengths. The fourth cervical is the longest. The trunk region is very crushed and not much can be observed. Limb elements bear slightly abraded surfaces, precluding observation of much detail. Scapulocoracoid, humeral epiphyses, and carpal elements are unfused. As preserved, metacarpal I reaches 82% the length of metacarpal IV, but its proximal end is unclear and it may have been longer. Both pedes are badly preserved and not much can be discerned.

Figure 9 New specimen D4019.

(A) Skeleton overview; (B) skull (left lateral view). (C and D) Respective schematic drawings. Abbreviations: ca, carpus; cv, cervical vertebra; d, dentary; d1–d4, digits 1–4; fpc, frontoparietala crest; h, humerus; j, jugal; m, maxilla; mc, metacarpal; mt, metatarsal; naof, nasoantorbital fenestra; pm, premaxilla; pmc, premaxillary crest; ph, phalanx; ti, tibia; ul, ulna; r, right; rad, radius; ri, rib; sk, skull. Scale bars: 50 mm.

Ontogenetic assessment

The new specimen D4019 is small-sized, with a humerus length of 64 mm (only slightly larger than the holotype of S. dongi, with a 58 mm humerus). It is too incomplete for a confident wingspan estimate. Based on the lack of ossification between scapulocoracoid elements, humeral epiphyses, and carpal elements, this individual is inferred as a juvenile.

BPMC 103 (new specimen)

Morphological summary

This specimen includes an almost complete skull (exposed in left lateral view), incomplete cervical series (exposed in dorsal view), incomplete forelimbs, and incomplete hindlimbs (Fig. 10). The rostrum is slender and deflected ventrally at an angle of 20°. A slight ventrolateral tilt of the plane of exposure of the rostrum reveals that the occlusal surface is sulcate, sporting thick tomial edges that emarginate an elongate sagittal excavation. Slit-like neurovascular foramina pierce the lateral surface of the rostrum close to the tomial edge (unclear in the occlusal surface). The premaxillary crest is large and protrusive. The anterior margin is roughly perpendicular to the main dorsal margin of the rostrum, anterodorsally oriented, similar to H. benxiensis and H. corollatus, and thus seems to have been originally trapezoidal in shape. The posterodorsal edge of the premaxillary crest is damaged, but it seems to have been anteroposteriorly longer than dorsoventrally high. The proportions of the nasoantorbital fenestra are not directly clear due to the disarticulation of the posterodorsal margin (nasal and lacrimal), but can be estimated at around 2.5 based on its length and mid-height. The dentary symphysis accounts for roughly 55% of total mandibular length, and sports a dorsal eminence as well as a low ventral crest. The anterior symphyseal region is pierced by slit-like foramina close to the occlusal line. Although the forelimbs are incompletely preserved, a partial humerus and both wing fingers are completely preserved. Metacarpal I preserves a clear proximal end and extends for only about 40% the length of metacarpal IV. The fourth wing phalanx accounts only for 20% of the first wing phalanx length. Metatarsal II is the longest one.

Figure 10 New specimen BPMC 103.

(A) Skeleton overview; (B) skull (left lateral view); (C) metacarpus, distal region. (D–F) Respective schematic drawings. Abbreviations: cv, cervical vertebra; d, dentary; d1–d4, digits 1–4; fpc, frontoparietala crest; h, humerus; j, jugal; m, maxilla; mc, metacarpal; mt, metatarsal; naof, nasoantorbital fenestra; pm, premaxilla; pmc, premaxillary crest; ph, phalanx; ti, tibia; ul, ulna; r, right; rad, radius; ri, rib; sk, skull. Scale bars: D, 50 mm; E, 20 mm; F, 10 mm.

Ontogenetic assessment

This specimen lacks fusion of the posterior skull elements, humeral epiphyses, carpals, extensor tendon process of the first wing phalanx, and tarsal elements. Unfortunately, neither pectoral nor pelvic girdles are preserved. This specimen may be a juvenile or a an early subadult. It is roughly equivalent in body size to the holotypes of H. jii, H. corollatus, and H. benxiensis, with a humerus of 79 mm in length and a wingspan of 1,546 mm.

BPMC 104 (new specimen)

Morphological summary

This specimen includes most of the skeleton, including a premaxillomaxilla, an almost complete mandible, incomplete cervical and dorsal series, and almost complete fore and hindlimb elements (Fig. 11). The rostrum is relatively robustly built and ventrally deflected at an angle of 20°. The rostrum deflection point is located roughly beneath the anterior margin of the nasoantorbital fenestra, where a bulge also seems to indicate the presence of a slight lateral palatal expansion. The premaxillary crest is unfortunately incompletely preserved, but it extends anterior to the anterior margin of the nasoantorbital fenestra and its broad base suggests it was relatively large. Despite the incompleteness of the skull, the length of the nasoantorbital fenestra can be assessed based on the location of the remains of the base of the lacrimal process of the jugal. The height of the nasoantorbital fenestra was measured at its mid-length, to account for the typical position of its maximum height limit as seen in more complete specimens. In this way, the length/height ratio of the nasoantorbital fenestra of BPMC 104 can be estimated at roughly 2.3. The lacrimal process of the jugal is not preserved. The jagged dorsal skull margin is reminiscent of the conditions seen in Tupandactylus (Campos & Kellner, 1997; Frey, Martill & Buchy, 2003), suggesting it sported a soft tissue crest. The dentary exhibits a dorsal eminence as well as a low ventral crest. Cervical formula cannot be assessed. The sacral vertebrae (number unclear) are partially fused and bear intersacral fenestrae. The coracoid bears a large ventral flange. The extension of metacarpal I can be assessed due to the good preservation of its proximal tip, despite the loss of some of the diaphysis (Fig. 11G). It extends for 41% the length of metacarpal IV, similar to H. benxiensis. The first wing phalanx exhibits a single pneumatic foramen on its ventral surface. The fourth wing phalanx is relatively reduced, corresponding to 20% of first wing phalanx length. In the pelvic girdle, the medial margin of the postacetabular process is excavated by a fossa, similar to Tapejara wellnhoferi and Vectidraco daisymorrisae (Eck, Elgin & Frey, 2011; Naish, Simpson & Dyke, 2013). The neck of the postacetabular process is relatively thick and elongate, similar to Vectidraco daisymorrisae (Naish, Simpson & Dyke, 2013) and unlike the rather constricted condition seen in Tapejara wellnhoferi (Eck, Elgin & Frey, 2011) or short condition seen in Tupandactylus navigans (Beccari et al., 2021). The femoral head exhibits a thick neck, with no visible constriction in posterior view. The greater trochanter is well-developed, and a large pneumatic foramen is present near its base. The distal end of the femur is expanded. In lateral view, the femur bows posteriorly. Two (?femoral) unfused epiphyses are present near the proximal end of the tibia. In the pedes, metatarsal II is the longest one.

Figure 11 New specimen BPMC 104.

(A) Skeleton overview; (B) skull (left lateral view); (C) left pelvis; (D) right femur; (E) right metatarsus; (F) right pelvis; (G) left metacarpus. (H–N) Respective schematic drawings. Abbreviations: cv, cervical vertebra; d, dentary; d1–d4, digits 1–4; etp, extensor tendon process; fe, femur; h, humerus; il, illium; is, ischium; m, maxilla; mc, metacarpal; mt, metatarsal; naof, nasoantorbital fenestra; pfo, pneumatic foramen; pm, premaxilla; ph, phalanx; pp, prepubis; pu, pubis; pv, pelvis; ti, tibia; ul, ulna; r, right; sca, scapula; sk, skull; ti, tibia. Scale bars: H, 50 mm; I, 50 mm; J–N, 10 mm; G, 50 mm.

Ontogenetic assessment

This specimen exhibits fusion of the scapulacoracoid, pelvic girdle (with closed, but still faintly visible, sutures), and distal tarsals. The extensor tendon process of the wing phalanx is still unfused, as are the posterior skull elements, ?femoral epiphyses, and proximal tarsals. The specimen is thus interpreted as a subadult. Concerning body size, this is the second largest specimen in our sample, with a humerus of 100 mm in length and a wingspan of 2,124 mm. This is one of only three Jiufotang tapejarid specimens at the 2-m wingspan size-class, along with D2525 (see above) and BPMC 107 (see below).

BPMC 105 (new specimen)

Morphological summary

Despite being relatively complete, this specimen is badly preserved—most bones are jumbled together, and most bone surfaces are badly weathered or cracked beyond the point of bearing relevant anatomical details (Fig. 12). Notwithstanding, the outlines of some bones and structures still reveal some interesting data. The skull, exposed in left lateral view, exhibits a trapezoidal premaxillary crest that is conspicuously protrusive, higher than anteroposteriorly long. The shape of the rostrum and the configuration of its ventral deflection are unclear. The nasoantorbital fenestra is about 2.2 times as long as high. The orbit seems to have been piriform. The dentary symphysis bears a dorsal eminence and a low ventral crest. Measurements for visible limb bones are given in File S1 (Sheet 1), but not much further comparative information can be retrieved. Wing proportions closely match those of H. corollatus and H. benxiensis, with the fourth wing phalanx accounting for roughly 20% the length of the first wing phalanx. The second metatarsal is the longest. Not much further information can be assessed.

Figure 12 New specimen BPMC 105.

(A) Skeleton overview; (B) skull (left lateral view); (C) detail of right manus. (D–F) Respective schematic drawings. Abbreviations: ca, carpal; etp, extensor tendon process; h, humerus; j, jugal; l, left; mand, mandible; mc, metacarpal; naof, nasoantorbital fenestra; or, orbit; pm, premaxilla; pmc, premaxillary crest; ul, ulna; r, right; ra, radius; rap, retroarticular process; sca, scapula; sk, skull; ti, tibia. Scale bars: A, D, 50 mm; B, C, E, F, 20 mm.

Ontogenetic assessment

Unfused elements include posterior skull bones and humeral epiphyses, carpals, the extensor tendon process of the first wing phalanx, and the tibiotarsus. It is, in this way, compatible with the holotype of S. dongi in both ontogenetic correlates and also body size, with a humerus of 69 mm in length and a wingspan of 1,288 mm (File S1, Sheet 6). This specimen may thus be regarded as a juvenile.

BPMC 106 (new specimen)

Morphological summary

This small specimen preserves mainly a partial skull (missing the rostrum) and partial forelimbs other than partial cervical and dorsal series, although not much can be observed (Fig. 13). A triangular, dorsally oriented premaxillary crest is present, located anterior to the inferred anterior limit of the nasoantorbital fenestra, similarly to the holotype of H. atavismus and specimens IVPP V 23388 and D4019. The dorsal edge of the premaxillary crest, and of the posterior process of the premaxilla as well, is jagged (as in Tupandactylus spp.; see Frey, Martill & Buchy, 2003), indicating the potential presence of soft tissue extension. The proportions of the nasoantorbital fenestra are unclear due to the incompleteness of the rostrum and disarticulation between the premaxillomaxilla and the posterior skull region. The shape of the jugal indicates the base of the orbit was broad, implying the orbit was probably subquadrangular/subcircular in shape. The first metacarpal is quite elongate, reaching at least 95% the length of the wing metacarpal.

Figure 13 New specimen BPMC 106.

(A) Skeleton overview; (B) skull (right lateral view). (C and D) Respective schematic drawings. Abbreviations: art, articular; cv, cervical vertebra; h, humerus; j, jugal; mc, metacarpal; np, nasal process; pm, premaxilla; po, postorbital; pt, pteroid; ul, ulna; rad, radius; sca, scapula; sk, skull. Scale bars: A, C, 50 mm; B, D, 10 mm.

Ontogenetic assessment

Unfused elements include the posterior skull elements, humeral epiphyses and the carpals. In terms of size, this specimen is relatively small in the Jiufotang tapejarid sample, with a humerus length of 69 (close to specimen D4019, and intermediate between the holotypes of S. dongi and H. jii; File S1, Sheets 1, 6). It is too incomplete for a confident wingspan estimate. The specimen is interpreted as a juvenile.

BPMC 107 (new specimen)

Morphological summary

This specimen comprises an almost complete skeleton, despite exhibiting badly preserved bone surfaces (Fig. 14). The rostrum is relatively slender and exhibits only a faint, incipient premaxillary crest, very similar to the holotype of S. dongi. The nasoantorbital fenestra is very elongate, with an estimated length/height ratio of about 3. The orbital region is not preserved. The posterodorsal region of the skull exhibits a short frontoparietal crest. The dentary symphysis is exposed in ventral view. It exhibits a dentary crest, which is dorsoventrally crushed and thus appears as a crushed keel. The dentary symphysis accounts for about half of mandibular length. The posterior region of the symphysis is damaged. The left mandibular ramus is complete, including the articular region and the retroarticular process, allowing for estimation of the location of the quadratomandibular articulation in the skull despite the absence of a preserved quadrate (and hence allowing for a rough estimation of the proportions of the nasoantorbital fenestra). The cervical series is incompletely preserved, and the longest cervical vertebra cannot be assessed. The preserved wings exhibit morphology and proportions comparable to the holotype of Sinopterus dongi, although metacarpals I–III cannot be assessed (File S1, Sheet 3). The sternum exhibits a rounded posterior margin. Metatarsal I is slightly longer than metatarsal II.

Figure 14 New specimen BPMC 107.

(A) Skeleton overview; (B) skull (right lateral view); (C) left humerus; (D) sternum; (E) left foot; (F) right pelvis. (G–L) Respective schematic drawings. Abbreviations: art, articular; ca, carpus; cv, cervical vertebra; d, dentary; d1–d4, digits 1–4; dpc, deltopectoral crest; f, frontal; fe, femur; fpc, frontoparietal crest; h, humerus; il, illium; is, ischium; l, left; mc, metacarpal; mt, metatarsal; naof, nasoantorbital fenestra; pmc, premaxillary crest; ph, phalanx; poap, postacetabular process; pt, pteroid; pu, pubis; ti, tibia; ul, ulna; r, right; rad, radius; sca, scapula; st, sternum; uc, ulnar crest. Scale bars: A, B, G, H, 50 mm; I–K, 10 mm; L, 30 mm.

Ontogenetic assessment

Specimen BPMC 107 is the largest Jiufotang tapejarid specimen within the sample analyzed here, with a humerus length of 111 mm and a totalized wingspan of ~2.18 m (only slightly larger than D2525 and BPMC 104). It bears several signs of an advanced ontogenetic stage, including complete fusion of the scapulacoracoid, humeral epiphyses, syncarpals, extensor tendon process, tibiotarsus, and almost complete fusion of the pelvic elements. Only the posterior skull elements were probably not completely fused, given their absence (presumably derived from disarticulation, implying lack of complete fusion). This specimen thus may be an advanced subadult close to skeletal maturity.

SMA—correlations and allometry

Of our 21 tested variables, six turned out to be unrelated to absolute humeral length (proxy for body size), and are thus interpreted as independent of body size and not explainable by ontogenetic variation (rostrum deflection, nasoantorbital fenestra length/height ratio, orbit ventral angle, quadrate inclination, metacarpal I length, and wing phalanx 4 length). Of the 15 variables that were recovered as correlated to absolute humerus length, two are positively allometric (femur and tibia length, though both are near-isometric) and two are negatively allometric (rostral value and rostrum index). These are thus interpreted as size-dependent, and thus easily explainable by ontogenetic variation. The remaining 11 traits were recovered as isometrically related to humerus length (Table 2).

Table 2 Results of the SMA analyses.

x	y	p	R²	Slope	Intercept	SMA lower CI	SMA upper CI	Correlation	n	
humerus	rostrum index	0.037	0.399	−1.103	3.87	−3.683	−0.55	–	11	
	rostral value	0.017	0.484	−0.703	1.988	−1.078	−0.36	–	11	
	rostrum def.	0.079	0.253	−0.13	0.722	0.22	1.09	N/C	13	
	naof l/h	0.172	0.195	−0.577	1.472	−2.299	−0.184	N/C	11	
	orbit°	0.392	0.248	−0.715	3.202	−3.437	0.146	N/C	5	
	Q°	0.231	0.591	−0.194	2.543	−0.514	0.003	N/C	4	
	cv IV	0.0056	0.746	0.872	−0.19	0.605	1.324	N/C	8	
	cv V	<0.001	0.855	1.047	−0.555	0.668	1.471	=	8	
	ul	<0.001	0.961	1.106	−0.029	0.957	1.238	=	16	
	pt	<0.001	0.864	1.122	−0.396	0.738	1.394	=	11	
	mcI	0.837	0.004	−1.89	5.479	−6.655	−0.905	N/C	12	
	mcIV	<0.001	0.922	1.1562	−0.08	0.933	1.372	=	16	
	ph1	<0.001	0.971	1.083	0.15	0.939	1.197	=	15	
	ph2	<0.001	0.965	1.081	0.034	0.947	1.201	=	15	
	ph3	<0.001	0.901	0.986	0.047	0.827	1.111	=	13	
	ph4	0.122	0.202	1.0535	−0.367	0.379	3.381	N/C	13	
	fe	<0.001	0.957	1.144	−0.179	1.00	1.262	≅ (+)	13	
	ti	<0.001	0.943	1.1811	−0.084	1.008	1.305	≅ (+)	15	
	mtI	<0.001	0.94	0.921	−0.249	0.701	1.131	=	11	
	mtII	<0.001	0.936	0.889	−0.174	0.616	1.081	=	12	
Note:

Abbreviations: CI, one-tailed 95% confidence interval; N/C, non-correlated; −, negative allometry; =, isometry; ≅, near-isometry; +, positive allometry. Anatomical abbreviations: cv, cervical vertebra; def., deflection; fe, femur; h, height/ l, length; mc, metacarpal; mt, metatarsal; naof, nasoantorbital fenestra; ph, wing phalanx; pt, pteroid; ti, tíbia; ul, ulna.

Linear morphometric multivariate analyses

As mentioned above (see Materials and Methods), our multivariate analyses only include skeletal proportions based on features which are interpreted as devoid of ontogenetic bias, i.e., features that are either uncorrelated to body size variation, or features that develop isometrically, as per the results of our SMA analyses. These features were rostrum deflection angle, nasoantorbital length/height ratio, ventral orbital angle, quadrate reclination angle, cervicals IV/V ratio, and the ratios between humerus and the following limb elements: ulna, pteroid, metacarpal I, metacarpal IV, wing phalanges 1–4, femur, metatarsal I, and metatarsal II (see File S1, Sheet 2, for the dataset). Skeletal proportions based on elements influenced by allometric development (rostrum index, rostral value, and tibia/humerus ratio) were not included. All analyzed taxa were included (i.e., sinopterines, Tapejara, Caiuajara, and Tupandactylus).

Under the results of our UPGMA analysis, members of the Sinopterus complex are segregated, distributed within two separate groups (Fig. 15). The first group, hereby termed Morphotype I, includes eight specimens: the holotypes of Sinopterus dongi, Huaxiapterus jii, Sinopterus lingyuanensis, and Huaxiapterus atavismus, as well as specimens IVPP V 23388, D4019, D7302, and BPMC 107. Morphotype I is the sister-cluster of Eopteranodon lii. The second morphotype, hereby termed Morphotype II, comprehends the remaining six analyzed specimens of the Sinopterus complex: the holotypes of H. corollatus and H. benxiensis, along with specimens D2525, BPMC 103, BPMC 104, and BPMC 105. Morphotype II is the sister-cluster to Tupandactylus navigans. Another cluster includes Tapejara wellnhoferi and Caiuajara dobruskii, as the sister-cluster to the whole remaining sample.

Figure 15 Resulting dendrogram of the UPGMA analysis.

Species names represent their holotypic specimens. Red indicates Morphotype I. Blue indicates Morphotype II.

Subsequent to our clustering analysis, PCA analyses were also carried out (Fig. 16). When groups corresponding to the UPGMA’s Morphotypes I and II are plotted onto the results of both of our PCA analyses, the resulting graphs reveal that there is no intersection between their convex hulls or 95% confidence ellipses. In the first PCA (using mean value imputation for missing data), there is an overlap between Eopteranodon lii and the 95% confidence ellipse of Morphotype I. Tapejara wellnhoferi, Caiuajara dobruskii and Tupandactylus navigans all fall outside of either morphotype’s morphospace. In the second PCA (using iterative imputation for missing data), a similar pattern is observed, except that Tupandactylus navigans falls within the 95% confidence ellipse of Morphotype II.

Figure 16 Results of the PCA analyses.

(A) Graph based on components 1 and 2 of the analysis using mean values imputation for the missing data. (B) Graph based on components 1 and 2 of the analysis using iterative imputation for the missing data. Species names represent their holotypic specimens. Red indicates Morphotype I (polygon represents convex hull; ellipse represents the 95% confidence ellipse). Blue indicates Morphotype II (polygon represents convex hull; ellipse represents the 95% confidence ellipse).

Kruskal-Wallis analyses

Kruskal-Wallis analyses were performed for each morphometric variable used in the multivariate analyses, except for quadrate inclination and cervical IV/cervical V ratio due to the low n available for Morphotype II (Table 3). Of the 13 analyzed variables, five turned out to reflect a statistically significant difference between Morphotypes I and II: rostrum deflection, nasoantorbital fenestra height/length, orbit ventral angle, metacarpal I/humerus length ratio, and wing phalanx 4/humerus length ratio (Table 3). The other analyzed variables do not show any significant difference between Morphotypes I and II.

Table 3 Results of the Kruskal-Wallis analyses.

Variable	n for M. I	n for M. II	H (chi2)	Hc	p	Significant difference	
rostrum def.	6	6	8.308	8.932	0.0028	Yes	
naof l/h	5	5	6.818	7.258	0.0070	Yes	
orbit°	3	3	3.857	4.091	0.0431	Yes	
Q°	3	1	–	–	–	–	
cvIV/cvV	6	1	–	–	–	–	
ul/hu	8	6	0.016	0.016	0.8973	No	
pt/hu	6	3	1.067	1.067	0.3017	No	
mcIV/hu	8	6	1.35	1.35	0.2453	No	
mcI/hu	6	5	7.5	7.5	0.0061	Yes	
ph1d4/hu	7	6	1	1	0.3173	No	
ph2d4/hu	7	6	0.510	0.510	0.4751	No	
ph3d4/hu	6	6	2.564	2.564	0.1093	No	
ph4d4/hu	6	6	8.308	8.308	0.0039	Yes	
mtI/hu	6	3	1.067	1.067	0.3017	No	
mtII/hu	7	3	0.6364	0.6364	0.4250	No	
Note:

Abbreviations: Anatomical abbreviations: cv, cervical vertebra; def., deflection; h, height/l, length; mc, metacarpal; mt, metatarsal; naof, nasoantorbital fenestra; ph, wing phalanx; pt, pteroid; ul, ulna.

Discussion

Anatomical variations and their interpretations (excluding cranial crests)

Based on the specimen-level remarks presented above, we discuss below the anatomical variations surveyed here for the Sinopterus complex. Our aim is to (1) identify and contextualize variation at specimen and morphotype levels, and (2) interpret these variations as potentially: sexual, ontogenetic, individual, or interspecific in nature. Of particular interest in this discussion are the features that, according to the results of our SMA analyses, are not correlated to body size and thus interpreted as not ontogenetic in nature; as well as the features that show significant variation between the two proposed morphotypes according to our Kruskal-Wallis analyses. Morphometric values shown to lack significant variation between the two proposed morphotypes are regarded as features that do not vary between the two morphotypes, and are thus set aside. For now, cranial crest variation will also be set aside, and addressed only further below, in order to circumvent the fact that these cannot be regarded a priori as a reliable source of either intra or interspecific variation.

It is worth highlighting that the amount of anatomical variation we were able to compilate here is, in a certain way, rather low if one considers that our sample includes several skeletons with high degrees of completeness. However, it must be observed that, unfortunately, such completeness is deceptive. The amount of information retrievable from these specimens is highly limited due to preservational issues. All the specimens are crushed and preserved in two-dimensions, so that in each specimen every bone is only visible from a single view. Some bones are further obliterated by other bones overlying them. Plus, some of these specimens also exhibit highly worn bone surface, precluding observation of many details (e.g., metatarsal lengths in D4019).

Rostrum, proportions (RI and RV)

Some variation in rostrum proportions in the Sinopterus complex had already been noted by Zhang et al. (2019). RI values vary from 2.85 (Sinopterus lingyuanensis) to 1.33 (BPMC 104), and RV values range from 6.53 (Huaxiapterus atavismus) to 3.5 (BPMC 104). It is clear that smaller, younger specimens tend to exhibit slenderer rostra, while larger, more mature specimens exhibit stouter rostra. In the present work, our SMA analysis indicates that both RI and RV are negatively allometric relative to body size. Since the measurement of RI is directly influenced by the presence and development of premaxillary crests, this result indicates that premaxillary crests grow with ontogeny in the Sinopterus complex. This result was to be expected, as cranial crest development is already well-known as an ontogenetic feature in pterosaurs, as demonstrated by taxa such as Caiuajara dobruskii (Manzig et al., 2014), Anhanguera spp. (Pinheiro & Rodrigues, 2017), and Pteranodon spp. (Bennett, 1993).

It is clear, as already noted by Witton (2013) and Naish, Witton & Martin-Silverstone (2021), that cranial crest development (and by extension, RI) should not be utilized as a tool for diagnosing potential Sinopterus complex species. Furthermore, we add here that the same is true for rostrum measurements that disregard crest development (i.e., RV), which show that the rostrum itself (exclusive of cranial crests) also develop to be stouter with ontogeny in the Sinopterus complex.

Rostrum, deflection angle

The ventral deflection of the rostrum is one of the most conspicuous cranial features of tapejarids, as seen in Caupedactylus, Tapejara, Tupandactylus, Caiuajara, Afrotapejara, Wightia, Eopteranodon, and in the Sinopterus complex. In the Sinopterus complex, Morphotype I exhibits a deflection angle range of 12°–15°, whereas the range is 20°–21° in Morphotype II (File S1, Sheet 5). Our SMA analysis indicates that this variation is not correlated to body size, and is thus interpreted as not ontogenetic in nature.

Martill et al. (2020a) had already reported on the intra- and interspecific variation of this feature within tapejarids. According to our own observations, deflection angles (as measured between the long axis of the deflected rostrum and the long axis of the maxilojugal bar) vary between 21°–25° in Tupandactylus imperator, 23°–25° in Tup. navigans, 25°–28° in Tapejara wellnhoferi, and 32°–37° in Caiuajara dobruskii (File S1, Sheet 5). We were unable to reproduce the measurements given by Martill et al. (2020a) for Caiuajara dobruskii, which produced a remarkably larger variation spectrum. This is probably explained by the variation in the shape of the palatal expansion bulge in Caiuajara dobruskii, which hampers the accurate measurement of the deflection angle if the maxillojugal bar posterior to it is not preserved (L. Piazentin, 2022, personal communication), and thus measurements of deflection angles in Caiuajara should be restricted to specimens with relatively complete maxillojugal bars. However, intraspecific variation in rostrum deflection in tapejarids does not seem to surpass a ~5° range.

Given the patterns of variation in other tapejarid species (File S1, Sheet 5), as well as the pattern that, within the Jiufotang sample, angles of 20°–21° are exclusive to Morphotype II while the other morphotypes are restricted to angles of 12°–15° (a difference which is statistically significant as indicated by our Kruskal-Wallis analysis), and that this variation is not related to body size, we regard this as a potential taxonomic signal for the Sinopterus complex.

Nasoantorbital fenestra, length/height ratio

Within the Jehol tapejarid sample, Morphotype I stands out due to its relatively elongate nasoantorbital fenestra, which is typically ~3 times as long as high (File S1, Sheet 1), as seen in the holotypes of S. dongi, S. jii, S. gui and S. lingyuanensis, and as can be roughly inferred from the holotype of S. atavismus as well as specimen IVPP 23388V. This contrasts with Morphotype II, in which the range for the nasoantorbital fenestra length/height ratio is 2.1–2.5. This seems to be roughly the typical condition for tapejarids, as seen in Eopteranodon lii (~2.1), Caupedactylus ybaka (~2.1), Tupandactylus imperator (∼2.5), and Tupandactylus navigans (2.1–2.2). Caiuajara and Tapejara stand out in exhibiting a relatively low ratio (~1.3; see File S1, Sheet 1). In this way, the particularly elongate nasoantorbital fenestra in Morphotype I is highly distinctive. Our SMA analysis indicates that nasoantorbital fenestra length/height ratio is not correlated to body size, and is thus interpreted as not ontogenetic in nature. It is worth highlighting that the elongate nasoantorbital fenestra of Morphotype I can be found in all skulls attributed to this morphotype, irrespective of ontogenetic stages—from the small juvenile holotype of S. gui to the large, subadult holotype of S. jii. Furthermore, our Kruskal-Wallis analysis indicate a significant difference between Morphotypes I and II regarding this feature. This feature could thus suggest a taxonomic distinction between Morphotype I and other tapejarids.

Nasal process, shape

In the holotypes of S. dongi and S. lingyuanensis (Morphotype I), the descending nasal processes are long, subvertically oriented, and extremely thin. This condition is also present in Tupandactylus navigans (Frey, Martill & Buchy, 2003). In Thalassodromeus sethi, the nasal process is also subvertical, although it is extremely reduced instead of elongated (Pêgas, Costa & Kellner, 2018). In the holotype of H. benxiensis (Morphotype II), however, the nasal process is anteriorly oriented, as already pointed out before in the data matrix of Andres, Clark & Xu (2014), although this feature was not explicitly reported in the original description of the specimen (Lü et al., 2007). Despite the substandard preservation of the nasal bones in the holotype of H. benxiensis (BXGM V0011), it can be seen upon close inspection that the right nasal, seen in medial view and partially overlayed by the incomplete left nasal, displays a preserved nasal process that is anteriorly inclined. Thus, we corroborate the coding provided by Andres, Clark & Xu (2014) and regard this very unusual condition as unique and possibly of taxonomic value for Morphotype II.

Orbit shape

Within the Sinopterus complex, variation exists concerning orbit shape, with the orbit being piriform in some specimens and subquadrangular/subcircular in others. This variation can be translated into the measurement of the angle between the lacrimal and postorbital processes of the jugal (ventral orbit angle), whereby subquadrangular orbits exhibit an angle close to 90° while piriform orbits exhibit a lower, acute angle. This difference in orbit shape distinguishes Morphotypes I and II. Morphotype I exhibits orbit angles of ~90°, as seen in the holotypes of S. dongi and S. lingyuanensis, as well as in IVPP 23388 V and BPMC SC04. Morphotype II exhibits lower values, with angles of 68° in the holotype of H. benxiensis and 65° in BPMC SC003. Within tapejarids, the peculiar subquadrangular orbit shape seen in Morphotype I is unique to this morphotype and to Eopteranodon lii (see below), with the piriform condition being the typical one, as seen in Morphotype II, Tapejara wellnhoferi, Tupandactylus navigans, Tupandactylus imperator, Caiuajara dobruskii, and Caupedactylus ybaka, as well as in thalassodromids (e.g., Pêgas, Costa & Kellner, 2018) and chaoyangopterids (e.g., Wu, Zhou & Andres, 2017). Our SMA analysis indicates that this feature is not correlated to body size. In effect, in Morphotype I, a perpendicular orbit angle can be found in both juveniles (holotypes of S. dongi and S. lingyuanensis, and specimen BPMC 106) and advanced subadults (IVPP 23388 V and holotype of H. jii). Our Kruskal-Wallis analysis indicate a significant difference between Morphotypes I and II regarding this feature. In this way, we regard that the distinctive orbit shape of Morphotype I most likely indicates distinctiveness between Morphotypes I and II.

Quadrate reclination

Within tapejarids, quadrate reclination (as measured between the maxillary ramus of the jugal and the quadrate) is usually between 140°–150°, as seen in Tapejara wellnhoferi (145°), Caiuajara dobruskii (147°), Tupandactylus navigans (145°), and Caupedactylus ybaka (150°). The holotype of Huaxiapterus benxiensis falls within this pattern, with a quadrate reclination of 147°. However, Morphotype I specimens exhibit a stronger quadrate reclination of 160°–162°, as seen in the holotypes of S. dongi and S. lingyuanensis, and specimens D4019 and BPMC 104 (File S1, Sheet 1). According to our SMA analysis, this variation is uncorrelated to body size. Quadrate reclination has been regarded as of taxonomic importance for Pteranodon (Bennett, 1994), tapejarids (Kellner, 2013), Nurhachius (Zhou et al., 2019), and wukongopterids (Zhou et al., 2021). Based on pterosaur species known from multiple specimens, intraspecific variation in quadrate reclination seems to surround 3°–6°, as seen in Pterodactylus antiquus, Aerodactylus scolopaciceps, Hamipterus tianshanensis, Pteranodon longiceps, and Pteranodon sternbergii (see Zhou et al., 2019). Although we do expect some minor influence of preservational distortion over this feature, this influence should also be accounted for in the 3°–6° variation pattern seen in the aforementioned species, as, similar to Jiufotang tapejarids, these are also represented by crushed specimens preserved on slabs as in the cases of Pterodactylus, Aerodactylus, and Pteranodon (e.g., Bennett, 1994). We thus regard this variation in the Sinopterus complex, which surpasses 10°, as of potential taxonomic value.

Cervical formula

Concerning the relative lengths of cervical vertebrae, the typical pterodactyloid condition is that the fifth is the longest one. This can be observed in chaoyangopterids (Leal et al., 2018; Wu, Zhou & Andres, 2017), azhdarchids (Naish & Witton, 2017), and Tupuxuara (Shen et al., 2021). However, some variation exists within tapejarids. In Tapejara wellnhoferi, cervicals four and five are roughly the same size (Vila Nova et al., 2015). In contrast, in Tupandactylus navigans the four cervical is longer than the fifth (Beccari et al., 2021). The same is true for Eopteranodon lii (Lü et al., 2006c) and for Morphotype I of the Sinopterus complex, as seen in the holotypes of S. dongi, S. lingyuanensis, specimen IVPP V 23388, and D3072. In contrast, the fourth cervical is shorter than the fifth in Morphotype II, as seen in the holotype of H. benxiensis (Lü et al., 2007). Our SMA analysis indicates that variation in cervical IV length is uncorrelated to body size in the Sinopterus complex. We thus regard this feature as of potential taxonomic value, distinguishing Morphotypes I and II.

Metacarpal I, length relative to metacarpal IV

An elongate metacarpal I that reaches the carpus is the plesiomorphic condition for azhdarchoids, as seen in Tapejara wellnhoferi, Eopteranodon lii, Tupuxuara leonardii, chaoyangopterids and azhdarchids (e.g., Kellner & Campos, 2007; Wu, Zhou & Andres, 2017). An interesting amount of variation regarding this feature has already been reported before for the Jiufotang tapejarids, with some specimens reportedly bearing either elongate metacarpals I that were subequal to (~90–100% the length of) metacarpal IV (e.g., Wang & Zhou, 2003) or reduced metacarpals I (e.g., Lü et al., 2006a).

Recently, Shen et al. (2021) expressed concern and recommended caution regarding this variation in the Sinopterus complex, since broken/obscured metacarpals could be mistaken for reduced metacarpals in some specimens. In the present work, our close inspection corroborates the presence of reduced metacarpals I (about 30–40% of the length of metacarpal IV) in specimen D2525 and in the holotypes of “H.” benxiensis and “H.” corollatus, along with the new specimens BPMC 103 and BPMC 104. Similarly, elongate metacarpals I (about 90–100% the length of metacarpal IV) are confirmed for the holotypes of S. dongi and H. jii, as well as specimens IVPP V 23388 and D3072, and the new specimen BPMC 106. The condition is unclear in the holotypes of S. lingyuanensis and “H.” atavismus.

Our SMA analysis indicates that this feature is uncorrelated to body size, and our Kruskal-Wallis analysis indicates a significant difference between Morphotypes I and II regarding this feature. This large amount of variation is unreported for pterosaur species and is highly suggestive that more than one species is present in this sample. Thus, we conclude that Morphotype II can be characterized by a short metacarpal I (about 30–40% of metacarpal I length), which does not reach the carpus. Such condition is unique for Morphotype II and Tupandactylus within all known tapejarids.

Wing digit, fourth phalanx length

Similar to what has been found for Rhamphorhynchus (Hone et al., 2021) and anurognathids (Yang et al., 2022), our SMA analyses indicate that most wing elements exhibit isometric growth within the Sinopterus complex. The sole exception to this pattern concerns fourth wing phalanx length, as our SMA analysis shows that its variation is not correlated to body size.

In fact, a noticeable variation occurs in this feature between Morphotypes I and II, irrespective of ontogenetic stage. The length ratio between wing phalanges 4 and 1 is about 0.30–0.40 in Morphotype I, as seen in the holotypes of S. dongi, S. jii, and “H.” atavismus, and specimen D3702. This is similar to Tapejara wellnhoferi and Caiuajara dobruskii, while in Eopteranodon lii the ratio is 0.45. In contrast, this ratio is no higher than 0.20 in Morphotype II, as seen in the holotypes of “H.” corollatus and “H.” benxiensis, as well as specimens D2525, BPMC SC001, BPMC SC002, and BPMC SC003. This is similar to Tupandactylus navigans, for which the same ratio is about 0.13 (Beccari et al., 2021). As indicated by our Kruskal-Wallis analysis, the difference between Morphotypes I and II regarding this feature is statistically significant in our sample. We thus regard that the short wing phalanx 4 (under 20% the length of wing phalanx 1, or under 50% the length of the humerus) of Morphotype II suggests taxonomic distinction from Morphotype I (and all other tapejarids, representing a potential diagnostic apomorphy).

Metatarsal I, relative length

The relative proportions of the metatarsals have already been deemed of taxonomic importance before (Zhang et al., 2019). The general tapejaroid condition is that metatarsal I is shorter than metatarsals II–III, and subequal to metatarsal IV; as found in Tapejara (Eck, Elgin & Frey, 2011), Eopteranodon (Lü et al., 2006c), chaoyangopterids (Wu, Zhou & Andres, 2017), and dsungaripterids (Hone, Jiang & Xu, 2018). This general condition can be seen in Morphotype II specimen D2525, although unclear in the holotypes of “H.” corollatus and “H.” benxiensis. On the other hand, the holotype of Sinopterus dongi and specimens D3702 and BPMC 106 are unique within tapejaroids in exhibiting an elongate metatarsal I (longer than metatarsal II). However, metatarsal I is subequal to metatarsal II (90–95% of its length) in other Morphotype I specimens, as seen in IVPP V 23388 (Zhang et al., 2019), and the holotypes of S. lingyuanensis (Fig. 7) and H. atavismus (Fig. 8), what is not significantly different from Morphotype II. Our SMA analysis indicates that metatarsals I and II grow isometrically relative to humeral length.

It is noticeable that an unusually long metatarsal I is exclusive to a subset of Morphotype I within the whole known tapejaroid sample, but, considering the data as a whole, it is difficult to set this subset of Morphotype I from the remainder of the morphotype, and thus this feature may only represent a polymorphism. Our Kruskal-Wallis analysis reveals no significant difference between Morphotypes I and II regarding this feature. We also note that this feature can be found elsewhere within pterosaurs, such as in Anurognathus ammoni (Bennett, 2007) and Anhanguera piscator (R. Pêgas, 2019, personal observation).

Species-level taxonomic interpretations

The primary taxonomic assessment presented in this subsection is based on the interpretation of the variations explored above (that is, excluding cranial crest variation). Each morphotype exhibits notorious, unique features, even when cranial crests are set aside. These particular features are summarized in Table 4 and Fig. 17.

Table 4 Summary of main anatomical variations surveyed here in Jehol tapejarids.

Species	Pmc shape	Rostrum def.	Naof l/h	Nasal process	Orbit shape	Q°	Longest cervical	Pt/ul	McI/ McIV	Ph4d4/ ph1d4	
Sinopterus dongi	Heaped	12°–15°	>3	Subv.	Subq.	~160°	cv IV	<0.50	>0.90	0.30–0.40	
Eopteranodon lii	Heaped	~15°	∼2.5	Subv.	Subq.	~160°	cv IV	>0.50	>0.90	~0.45	
Huaxiadraco corollatus	Subq.	~20°	2.2–2.5	Ant.	Pirif.	~150°	cv V	~0.50	0.30–0.40	~0.20	
Note:

Abbreviations: ant., anteriorly directed; cv, cervical; d, digit; def., deflection; h, height; l, length; mc, metacarpal; ph, phalanx; pirif., piriform; pmc, premaxillary crest; pt, pteroid; Q°, quadrate inclination; ul, ulna; subq., subquadrangular; subv., subvertical.

Figure 17 Anatomical variation in the skull and wing in the Sinopterus complex.

(A) Schematic representation of skull and metacarpus + wing digit of Morphotype I (S. dongi). Based mainly on the holotype of S. dongi (IVPP V 13363), except for the premaxillary crest (based on the holotype of H. jii; GMN-03-11-001). (B) Schematic representation of skull and metacarpus + wing digit of Morphotype II (H. corollatus). Based mainly on the holotype of H. benxiensis (BXGM V0011). (A) and (B) are not to scale, but are both proportionately scaled to the same metacarpal IV length. Numbers indicate features that vary between S. dongi and H. corollatus (see text): (1) orbit shape; (2) quadrate reclination; (3) nasal process shape; (4) nasoantorbital fenestra length/height ratio; (5), premaxillary crest shape; (6) rostrum deflection; (7) metacarpal I length; (8) wing phalanx 4 length.

Morphotypes I and II are quite distinguishable from each other. Within the Jiufotang tapejarid sample, Morphotype I is characterized by a subquadrangular orbit, a gentle rostrum deflection of 12°–15°, an elongate nasoantorbital fenestra (over three times as long as high), a subvertical nasal process, a quadrate reclination of ~160°, a fourth cervical longer than the fifth, and an elongate wing phalanx 4 (about 30–40% the length of the first wing phalanx). Morphotype II differs from Morphotype I in exhibiting a piriform orbit, a stronger rostrum deflection of 20°–22°, a stouter nasoantorbital fenestra (about 2.2–2.5 times as long as high), a nasal descending process anteriorly oriented, a quadrate reclination of ~150°, a fourth cervical shorter than the fifth, a reduced metacarpal I far from contacting the carpus (30–40% the length of metacarpal IV), and a short wing phalanx 4 (about 20% the length of the first wing phalanx).

We regard that these different combinations of features cannot be attributed to ontogenetic variation, as indicated by our SMA analyses—all of the characteristics mentioned above are uncorrelated to body size (see above). These features also fail to match what (little) is known about sexual dimorphism in pterosaurs (see Bennett, 1993, Wang et al., 2014). This great amount of variation also surpasses the level of individual variation that is seen in the few known monospecific pterosaur bonebeds (see Manzig et al., 2014; Wang et al., 2014; Andres & Langston, 2021). Furthermore, it is notorious that these variations are consistently co-occurrent, effectively allowing us to segregate two morphotypes without overlap (each with their own unique features), what is suggestive of heterobatmy. We thus regard that these features are best interpreted as interspecific in nature. Based on the weight of these combined features, we regard that each morphotype is, indeed, distinct from each other at the species-level, meaning the Jiufotang tapejarid sample would comprise at least two species.

As we are unable to satisfactorily distinguish proposed species within each morphotype, we interpret that a single species is present in each morphotype. In this way, Morphotype I would represent Sinopterus dongi, with S. gui, H. jii, S. lingyuanensis, and H. atavismus as junior synonyms. From heretofore, ‘Huaxiapterus’ will thus be referred to between single quotation marks to indicate its status as invalid (as a subjective junior synonym of Sinopterus). Morphotype II would represent ‘H.’ corollatus, with ‘H.’ benxiensis as a junior synonym. A reinterpretation of the taxonomic attribution of Jiufotang tapejarid specimens, based on the aforementioned remarks, is presented in Table 5.

Table 5 Summary of taxonomic attributions of Sinopterus complex specimens.

Specimen	Original attribution	Reference	Present attribution (this work)	
Sinopterus dongi holotype	Sinopterus dongi	Wang & Zhou (2003)	Sinopterus dongi	
Sinopterus gui holotype	Sinopterus gui	Li, Lü & Zhang (2003)	Sinopterus dongi	
Huaxiapterus jii holotype	Huaxiapterus jii	Lü & Yuan (2005)	Sinopterus dongi	
Huaxiapterus corollatus holotype	Huaxiapterus corollatus	Lü et al. (2006a)	Huaxiadraco corollatus	
D2525	Sinopterus dongi	Lü et al. (2006b)	Huaxiadraco corollatus	
Huaxiapterus benxiensis holotype	Huaxiapterus benxiensis	Lü et al. (2007)	Huaxiadraco corollatus	
Nemicolopterus crypticus holotype	Nemicolopterus crypticus	Wang et al. (2008)	Sinopterinae indet.	
PMOL-AP00030	Tapejaridae indet.	Liu et al. (2014)	Sinopterinae indet.	
Sinopterus lingyuanensis holotype	Sinopterus lingyuanensis	Lü et al. (2016)	Sinopterus dongi	
Huaxiapterus atavismus holotype	Huaxiapterus atavismus	Lü et al. (2016)	Sinopterus dongi	
IVPP V 23388	Sinopterus atavismus	Zhang et al. (2019)	Sinopterus dongi	
D3072	Sinopterus dongi	Shen et al. (2021)	Sinopterus dongi	
SDUST-V1012	Sinopterus sp.	Zhou, Niu & Yu (2022)	Sinopterinae indet.	
SDUST-V1014	Sinopterus sp.	Zhou et al. (2022)	cf. Sinopterus dongi	
D4019	–	–	Sinopterus dongi	
BPMC 103	–	–	Huaxiadraco corollatus	
BPMC 104	–	–	Huaxiadraco corollatus	
BPMC 105	–	–	Huaxiadraco corollatus	
BPMC 106	–	–	Sinopterus dongi	
BPMC 107	–	–	Sinopterus dongi	

We agree with Witton (2013) and Naish, Witton & Martin-Silverstone (2021) that, at the time of their writings, evidence for multiple Jiufotang tapejarid species was insufficient due to the lack of detailed data on their anatomical variation. Still, the present work provides new anatomical and comparative data which we interpret as compelling evidence for the existence of two tapejarid species in the Jiufotang Formation.

Overlap in the stratigraphic distributions of S. dongi and ‘H.’ corollatus suggests that these two proposed species have coexisted in time. We emphasize that the occurrence of a few sympatric, closely related pterosaur species should not be viewed, by default, as unlikely. As observed by Longrich, Martill & Andres (2018), sympatry of closely related species is not uncommon for seabirds (e.g., species of Fregata, Larus), and we add here that the same is true for continental birds (e.g., species of Cathartes, Accipiter, Ramphastos, Ara, Amazona; e.g., Sigrist, 2004; Billerman et al., 2022). Despite stratigraphic overlap, Sinopterus dongi and ‘H.’ corollatus are not yet known from the exact same localities. Thus, the possibility remains that these two species took part in some sort of niche partitioning, as has been proposed for the two species of Quetzalcoatlus that co-occur in the layers of the Javelina Formation: while the giant Q. northropi has been recovered from stream channel facies, the remains of the smaller Q. lawsoni stem from abandoned channel-lake facies (Andres & Langston, 2021; Brown, Sagebiel & Andres, 2021; Lehman, 2021). Further work on the lithology and depositional environments of Jiufotang localities may shed light on this possibility for Jiufotang tapejarids as well.

Cranial crest variation: mapping and interpretation

Premaxillary crest, development and size

Within the entire sample of Jiufotang tapejarids here analyzed, only two specimens lack premaxillary crests: the holotypes of Sinopterus gui and S. lingyuanensis. These are the two smallest specimens analyzed here, and are both interpreted as juveniles. The holotype of Nemicolopterus crypticus, which is a near-hatchling, may represent a third specimen of crestless Jiufotang tapejarid (see Witton, 2013; Naish, Witton & Martin-Silverstone, 2021). Thus, it can be said that premaxillary-crestless Jiufotang tapejarids are restricted to very young individuals.

All remaining Jiufotang tapejarid specimens exhibit premaxillary crests, but of differing sizes. Within our Morphotype I (=Sinopterus dongi), the holotype of Sinopterus dongi exhibits but a very discrete crest; that is, it only discretely disrupts the skull margin (by protruding anterodorsally). This specimen is interpreted as a juvenile (see above). Specimen BPMC 107 exhibits a similarly incipient premaxillary crest, despite being regarded as close to skeletal maturity. In contrast, the holotype of ‘H.’ atavismus, which is regarded as a juvenile and is smaller in size than the holotype of S. dongi, bears a more conspicuous premaxillary crest than the latter specimen.

Because premaxillary-crestlessness is restricted to the smallest juvenile specimens, it seems clear that premaxillary crest absence/presence is an ontogenetic feature. Ontogenetic variation in the presence and development of premaxillary crests is corroborated by our SMA analysis, which indicates negative allometry between rostrum index and body size. The negatively allometric growth of rostrum index can easily be explained by the ontogenetic development of the premaxillary crest, which is a feature that augments the value of the rostrum index.

In addition, because premaxillary crest size still varies between larger juveniles and subadults of each species, it also seems likely that variation in premaxillary crest size is influenced also by individual and/or sexual variation, and not only to growth. These variations concerning premaxillary crest presence and size are clearly affected by intraspecific (ontogenetic, individual, and sexual) variations and seem to apply to the Sinopterus complex as a whole. Thus, these variations (concerning crest presence and size) should not be regarded as taxonomic informative for the Sinopterus complex. On the other hand, stating that crest presence/size cannot differentiate between species within our analyzed sample does not imply that a single species exists. Rather, it suggests that sexual and ontogenetic variation (expressed in premaxillary crest size/development) is present in all potentially valid species, whether a single one or more (two as we propose here).

Premaxillary crest, shape

Apart from premaxillary crest presence and size, variation in crest shape can also be seen in Jiufotang tapejarids. Crested specimens exhibit crests of roughly two shapes: heaped and trapezoidal.

Morphotype I is characterized by heaped crests, as seen in the holotypes of S. dongi, ‘H.’ jii, ‘H.’ atavismus, and specimens IVPP 23388 V, D4019, BPMC 106, and BPMC 107. In contrast, Morphotype II is characterized by trapezoidal crests, as seen in the holotypes of ‘H.’ corollatus and ‘H.’ benxiensis, as well as specimens BPMC 103 and BPMC 105.

It is important to highlight that we do not mean to imply that these proposed shape categories are homogenous. Some degree of variation is evidently present within each of them and no two crests are the same, as should be expected given the intraspecific variation in cranial ornamentation that is seen in extant vertebrates, such as in the casques of Numida (Angst et al., 2020) and Casuarius (Naish & Perron, 2016; Green, Kay & Gignac, 2022).

It is notorious that these two shape categories match the two recognized morphotypes/species, apparently without overlap or relation to ontogeny. Consequently, the favored explanation under this scenario is that each shape is characteristic of each morphotype/species.

Comments on the usage of cranial crests in pterosaur taxonomy

In summary, we interpret here that, within Jiufotang tapejarids, (1) variation in crest presence/development is linked to ontogeny, (2) variation in crest size can be also linked to individual/sexual variation, and (3) crest shape is linked to interspecific variation. As an example of a similar case, we can mention the Pteranodon complex. By following the most restrictive taxonomic interpretation of this species complex (Bennett, 1994; Martin-Silverstone et al., 2017), it can be said that crest shape (as seen in proposed mature males) is diagnostic for the two valid Pteranodon species: elongate and posteriorly oriented in Pteranodon longiceps, and “bulbous” and upright in Pteranodon sternbergi (Bennett, 1994). In contrast to that, juveniles and females of these two Pteranodon species cannot be set apart by cranial crest morphology, since these morphs would bear underdeveloped crest morphologies (Bennett, 1994; Martin-Silverstone et al., 2017). We regard that we should expect for pterosaurs the same amount of complexity we see in extant birds: species with and without sexual dimorphism in ornaments; closely related species with distinct (and diagnostic) ornaments; and closely related species with similar ornaments. We regard here that each case will need its own assessment, and that no general pattern should be expected for pterosaurs as a whole—a very diverse group that radiated for over 165 million years.

Short comments on Nemicolopterus crypticus

As observed by Witton (2013) and Naish, Witton & Martin-Silverstone (2021), the holotype specimen of Nemicolopterus crypticus (Fig. 18) clearly represents a young juvenile, as indicated by its “small size, proportionally enormous orbit, rounded and unfused pelvic bones, poorly defined limb articulations with unfused epiphyses, unfused skull bones, unfused scapulocoracoid, and lack of fusion between the tibia and tarsus” (Naish, Witton & Martin-Silverstone, 2021). Furthermore, it resembles tapejarids due to a combination of several features, most importantly edentulousness, a downturned rostrum, a slender and subvertical lacrimal process of the jugal, a jaw joint ventral to the anterior half of the orbit, and relatively elongate hindlimbs (Naish, Witton & Martin-Silverstone, 2021). We further note that one of the proposed diagnostic features of Nemicolopterus crypticus, a penultimate phalanx of pedal digit 4 longer than the first (Wang et al., 2008), is a feature it shares with Jiufotang tapejarids (e.g., Shen et al., 2021; Zhou et al., 2022).

Figure 18 Nemicolopterus crypticus holotype (IVPP V-14377).

(A) Skeleton overview, and (B) schematic drawing. (C) Skull (right lateral view), and (D) schematic drawing. Abbreviations: cdv, caudal vertebrae; co, coracoid; cv, cervical vertebra; d1–d4, digits 1–4; f, frontal; fe, femur; h, humerus; hy, hyoid; il, illium; is, ischium; j, jugal; l, left; la, lacrimal; mand, mandible; mc, metacarpal; mt, metatarsal; naof, nasoantorbital fenestra; or, orbit; pa, parietal; pm, premaxilla; ph, phalanx; ti, tibia; r, right; sca, scapula. Scale bars: A–B, 100 mm; C–D, 5 mm.

Here, we highlight that Nemicolopterus crypticus exhibits a morphology that is far distinct from any other Jiufotang tapejarid specimen, what can be attributed to its very young stage—this is expressed by the entire lack of cranial crests, a relatively large orbit, a relatively diminutive nasoantorbital fenestra, a not much reclined quadrate, and a “knife-shaped” humeral deltopectoral crest (Wang et al., 2008). Absence of cranial crests and large orbits are well-known indicators of young ontogenetic stages (e.g., Bennett, 1993). It is interesting to note that the distinctive shape of the humeral deltopectoral crest of the holotype of N. crypticus could easily be explained by an incipient ossification of the structure—in fact, neonate specimens of Hamipterus tianshanensis seem to be characterized by incipiently ossified humeral deltopectoral crests (Wang et al., 2017).

Concerning the holotype of N. crypticus, we regard that its very early juvenile status (near-hatchling; Naish, Witton & Martin-Silverstone, 2021) is insufficient for a satisfactory diagnosis and prevents a confident identification as conspecific with either S. dongi or ‘H.’ corollatus (or yet a distinct species). Thus, we consider that the holotype of Nemicolopterus crypticus should be regarded as an indeterminate Sinopterinae (Table 5).

Short comments on Eopteranodon lii

As discussed above, Eopteranodon lii is a tapejarid species that comes from the Yixian Formation, which is slightly older than and underlies the Jiufotang Formation (from which the Sinopterus complex comes from). Eopteranodon lii has been regarded as a close relative of the genus Sinopterus in several phylogenetic analyses (Vullo et al., 2012; Andres, Clark & Xu, 2014; Pêgas et al., 2021), a result that is corroborated here (see below). However, the tapejarid nature of Eopteranodon lii has not been consensual. This taxon has been, at times, interpreted as a chaoyangopterid (e.g., Lü et al., 2008). Furthermore, Martill et al. (2020b) noted that a tapejarid-like downturned rostrum could not be verified in the holotype of Eopteranodon lii due to the lack of detailed illustrations, and that a re-study of the holotype would be desirable. Close analysis of the type specimen reveals clear tapejarid features (Figs. 19 and 20), including a downturned rostrum with a premaxillary crest (note that the original identifications of skull and mandibular remains were mistakenly switched).

Figure 19 Eopteranodon lii holotype (BPV 078).

(A) Counterpart; (B) main part. (C and D) Respective schematic drawings. Abbreviations: cv, cervical vertebra; co, coracoid; d1–d4, digits 1–4; fe, femur; fi, fibula; h, humerus; j, jugal; mand, mandible; mc, metacarpal; pmc, premaxillary crest; pe, pelvis; ph, phalanx; ti, tibia; ul, ulna; rad, radius. Scale bars: C, 50 mm; D, 10 mm.

Figure 20 Eopteranodon lii holotype (BPV 078) details.

(A) Close-up of the specimen’s main part, and (B) schematic drawing. Abbreviations: art, articular; d, dentary; h, humerus; m, maxilla; naof, nasoantorbital fenestra; pmc, premaxillary crest; ppm, posterior premaxillary process. Scale bars: 50 mm.

Eopteranodon lii exhibits striking similarities to Sinopterus dongi, especially in orbit shape (subquadrangular), quadrate reclination (about 160°), and in cervical IV being the longest one. Still, Eopteranodon lii differs from Sinopterus dongi in exhibiting a stouter nasoantorbital fenestra (about 2.5 times as long as high), a fairly elongate pteroid (pteroid/ulna length ratio about 0.56), an elongate wing phalanx 4 (wing phalanx 4/phalanx 1 length ratio about 0.45), and a metatarsal I shorter than metatarsal II. Thus, we corroborate the distinction between Eopteranodon lii and Sinopterus dongi, as well as ‘H.’ corollatus.

We further note that, due to the close proximity between Eopteranodon lii and Sinopterus dongi, and to the fact that the former is chronologically older than the latter, it is possible that Eopteranodon lii and Sinopterus dongi could be linked in an anagenetic continuum and thus represent chronospecies. This is similar to what has been proposed for other closely related pterosaur species that are stratigraphically successive: Pteranodon sternbergi and P. longiceps (Bennett, 1994), and Nurhachius luei and N. ignaciobritoi (Zhou et al., 2019).

Phylogenetic analysis

Our search produced three minimum-length trees, with 551 steps, ensemble consistency index of 0.593 and ensemble retention index of 0.860. In our strict consensus tree (Fig. 21), we recovered a clade of Jehol tapejarids, in which the clade Eopteranodon lii + Sinopterus dongi is the sister-group of ‘H.’ corollatus. This Jehol clade (comprising Eopteranodon lii, Sinopterus dongi, and ‘H.’ corollatus) is supported by the following unambiguous synapomorphies: char. 109(1) posteriorly shifted apex of the dentary dorsal eminence (located posterior to the anterior third of mandibular length); char. 127(2) concave dorsal margin of the mandibular ramus; and char. 131(2), elongate retroarticular process (char. 161 of Wu, Zhou & Andres, 2017).

Figure 21 Time-calibrated strict consensus tree.

The two species of the Sinopterus complex here regarded as valid are indicated in dark red. 1: Tapejaromorpha. 2: Thalassodromidae. 3: Tapejaridae. 4: Caupedactylia. 5: Tapejarinae. 6: Sinopterinae. 7: Azhdarchomorpha. 8: Chaoyangopteridae. 9: Alanqidae. 10: Azhdarchidae.

The node joining Eopteranodon lii and Sinopterus dongi was supported by the following four synapomorphies: char. 8(1), subquadrangular orbit; char. 30(0), skull height (from squamosal to premaxilla, exclusive of cranial crests) relative to jaw length under 25% of jaw length (modified from Witton, 2012; Andres, Clark & Xu, 2014); char. 70(4) quadrate reclination about 160° (ambiguous synapomorphy); and char. 178(1) fourth mid-cervical longer than the fifth.

Based on the compelling anatomical differences between S. dongi and ‘Huaxiapterus’ corollatus, along with the fact that S. dongi is recovered here as closer to E. lii than to ‘Huaxiapterus’ corollatus, we regard that ‘Huaxiapterus’ corollatus requires a new generic name—agreeing with previous suggestions (Kellner & Campos, 2007) and phylogenetic analyses (Andres, Clark & Xu, 2014). We thus erect Huaxiadraco gen. nov. to accommodate Huaxiadraco corollatus comb. nov. (Fig. 22).

Figure 22 Life reconstruction of the Jiufotang tapejarids.

The coexistence between Sinopterus dongi and Huaxiadraco corollatus comb. nov. in the Jiufotang paleoenvironment. Art: courtesy of Zhao Chuang.

It is interesting to note that the relationships between the Jehol tapejarid species as recovered by our phylogenetic analysis is different from the distance-based relationships between the morphotypes in our morphometric analysis. Particularly, Tupandactylus navigans is recovered closer to Morphotype II than to Tapejara wellnhoferi and Caiuajara dobruskii. It is important to bear in mind that the cluster analysis is based on similarity (which are measured by distance, and can reflect homoplasy), and not shared traits (as is the case of the phylogenetic analysis). This kind of analysis may produce useful information on a species-level taxonomy (granted the analyzed traits are not sexual or ontogenetic in nature, as discussed here), but it has no bearing on the phylogenetic relationships between the analyzed species. While our species circumscriptions are based on morphological and morphometric variation (thus the utility of a specimen-level phenogram in order to cluster specimens), our generic attributions must be guided by our phylogenetic results.

Systematic Paleontology

Pterosauria Owen, 1842

Pterodactyloidea Plieninger, 1901

Azhdarchoidea Unwin, 1995 (sensu Kellner, 2003)

Tapejaromorpha Andres, Clark & Xu, 2014 (sensu Andres, 2021)

Tapejaridae Kellner, 1989

Node-based definition. The least inclusive clade containing Tapejara wellnhoferi Kellner, 1989, Sinopterus dongi Wang & Zhou, 2003, and Caupedactylus ybaka Kellner, 2013 (unrestricted emendation). Reference phylogeny: Fig. 21.

Composition. Caupedactylia clad. nov. and Eutapejaria clad. nov. (see Table 1). Caupedactylia contains Caupedactylus ybaka and Aymberedactylus cearensis. Eutapejaria contains Tapejarinae and Sinopterinae (see below).

Diagnostic apomorphies. Lateral expansion of the jaws (both) level with anterior margin of the nasoantorbital fenestra; main part of dorsal skull margin (excluding cranial crests) convex in lateral view; prenarial rostrum and dentary symphysis ventrally deflected; lacrimal bearing extensive fenestration; dentary symphysis bearing a ventral sagittal crest.

Remarks. The original PhyloCode-compliant phylogenetic definition (Andres, 2021) is unrestrictedly emended here by the simple addition of Caupedactylus ybaka as a third internal specifier. Although this taxon was not included in the reference phylogeny from Andres (2021), it is recovered here as closely related to tapejarines and sinopterines (sensu Andres, 2021) as in previous studies (e.g., Vidovic & Martill, 2014; Pêgas et al., 2021), due to exhibiting a series of well-established diagnostic features of Tapejaridae (sensu Lü et al., 2006a; Andres, 2021), as listed above. The present unrestricted emendation is thus done to preserve the stability of Tapejaridae in terms of diagnosis, usage, and content, under the context of the present reference phylogeny (Fig. 21). Tapejaridae (sensu this work) thus includes Caupedactylia and Eutapejaria (see Table 1).

Eutapejaria new clade name

Branch-based definition. The most inclusive clade containing Tapejara wellnhoferi Kellner, 1989 but not Caupedactylus ybaka Kellner, 2013. Reference phylogeny: Fig. 21.

Composition. Tapejarinae (sensu Andres, 2021) and Sinopterinae (sensu Andres, 2021). Tapejarinae contains Tapejara wellnhoferi, Caiuajara dobruskii, Tupandactylus imperator, Tupandactylus navigans, and Europejara olcadesorum. Sinopterinae contains Sinopterus dongi, Eopteranodon lii, Huaxiadraco corollatus gen. et comb. nov., Bakonydraco galaczi, Afrotapejara zouhri, and Wightia declivirostris.

Diagnostic apomorphies. Marked gap between jaws during occlusion; premaxillary crest anteriorly tall and forming a low, rod-like process extending posteriorly; dorsal dentary eminence present on the dentary symphysis; dentary symphysis anterior surface sulcate with thick, well-marked tomial edges; humeral ulnar crest rounded in shape and posterodorsally flared.

Sinopterinae Lü et al., 2016 (sensu Andres, 2021)

Sinopterus dongi Wang & Zhou, 2003

Holotype. IVPP V 13363.

Referred material. BPV-077, GMN-03-11-001, JPM-2014-005, XHPM 1009, IVPP V 23388, D3072, D4019, BPMC 106, BPMC 107.

Synonymy. Sinopterus gui Li, Lü & Zhang (2003), Huaxiapterus jii Lü & Yuan (2005), Sinopterus lingyuanensis Lü et al. (2016), and Huaxiapterus atavismus Lü et al. (2016).

Type locality and horizon. Chaoyang City of Liaoning Province. Jiufotang Formation.

Diagnostic apomorphies. Sinopterinae with the following unique features (autapomorphies): nasoantorbital fenestra relatively elongate (over three times as long as high); pteroid shorter than half of ulna length; metatarsal I subequal to or longer than metatarsal II (longer than metatarsal III).

Differential diagnosis. Sinopterinae species with the following combination of features: premaxillary crest heaped in outline, in the crested morph (=Eopteranodon, ≠Huaxiadraco); rostrum deflection of 12–15° (=Eopteranodon, ≠Huaxiadraco); nasoantorbital fenestra relatively elongate, over three times as long as high (autapomorphy); nasal process subvertical and elongate (=Eopteranodon, ≠Huaxiadraco); subquadrangular orbit (=Eopteranodon, ≠Huaxiadraco); quadrate reclination of ~160° (=Eopteranodon, ≠Huaxiadraco); fourth cervical vertebrae the longest (=Eopteranodon, ≠Huaxiadraco); pteroid shorter than half of ulna length (autapomorphy); metacarpal I subequal to metacarpal IV (=Eopteranodon, ≠Huaxiadraco); wing phalanx 4/phalanx 1 length ratio about 0.30 (≠Eopteranodon, ≠Huaxiadraco); metatarsal I subequal to or longer than metatarsal II, and longer than metatarsal III (autapomorphy).

Eopteranodon lii Lü & Zhang, 2005

Holotype. BPV-078.

Referred material. D2526.

Type locality and horizon. Beipiao, Liaoning Provice. Yixian Formation.

Diagnostic apomorphies. Tapejarid with the following autapomorphies: elongate pteroid (pteroid/ulna length ratio about 0.56); elongate wing phalanx 4 (subequal to phalanx 3 and about 45% the length of phalanx 1).

Differential diagnosis. Sinopterinae with following combination of features: premaxillary crest heaped in outline, in the crested morph (=Sinopterus, ≠Huaxiadraco); rostrum deflection of 15° (=Sinopterus, ≠Huaxiadraco); nasoantorbital fenestra relatively stout, about 2.5 times as long as high (≠Sinopterus, =Huaxiadraco); nasal process subvertical and elongate (=Sinopterus, ≠Huaxiadraco); subquadrangular orbit (=Sinopterus, ≠Huaxiadraco); quadrate reclination of ~160° (=Sinopterus, ≠Huaxiadraco); fourth cervical vertebrae the longest (=Sinopterus, ≠Huaxiadraco); pteroid over half of ulna length (autapomorphy); metacarpal I subequal to metacarpal IV (=Sinopterus, ≠Huaxiadraco); elongate wing phalanx 4, subequal to phalanx 3 and about 45% the length of phalanx 1 (autapomorphy); metatarsal I shorter than metatarsal II (≠Sinopterus, =Huaxiadraco).

Huaxiadraco gen. nov.

Etymology. After Huaxia, an ancient, pre-imperial name for the Chinese civilization (literal meaning: beautiful grandeur), and draco, Latin for dragon.

Type species. Huaxiadraco corollatus (Lü et al., 2006a), new combination.

Diagnosis. As for type and only species.

Huaxiadraco corollatus (Lü et al., 2006a) comb. nov.

Holotype. ZMNH M813.

Referred material. BXGM V0011, D2525, BPMC 103, BPMC 104, BPMC 105.

Synonymy. Huaxiapterus benxiensis Lü et al. 2005.

Type locality and horizon. Chaoyang City of Liaoning Province. Jiufotang Formation.

Diagnostic apomorphies. Sinopterinae with the following unique features (autapomorphies): premaxillary crest trapezoidal in shape and slanting anterodorsally (in the crested morph); nasal descending process anteriorly oriented; short metacarpal I (30–40% the length of metacarpal IV); and short wing phalanx 4 (~20% the length of wing phalanx 1).

Differential diagnosis. Sinopterinae species with premaxillary crest trapezoidal in shape and slanting anterodorsally, in the crested morph (autapomorphy); orbit piriform in shape (≠Sinopterus, ≠Eopteranodon); rostrum deflection of ~20° (≠Sinopterus, ≠Eopteranodon); nasoantorbital fenestra relatively stout, 2.2–2.5 times as long as high (≠Sinopterus, =Eopteranodon); quadrate reclination of ~150° (≠Sinopterus, ≠Eopteranodon); fifth cervical vertebrae the longest (≠Sinopterus, ≠Eopteranodon); short metacarpal I, 30–40% the length of metacarpal IV (autapomorphy); short wing phalanx 4, ~20% the length of wing phalanx 1 (autapomorphy); and metatarsal I shorter than metatarsal II (≠Sinopterus, =Eopteranodon).

Conclusions

Jiufotang tapejarids were originally divided into seven nominal species, all entangled in a series of disputed interpretations. Our qualitative and quantitative reassessments lead us to recognize two morphotypes of Jiufotang tapejarids, and to conclude that each of these morphotypes represents a distinct species. These are: Morphotype I, corresponding to Sinopterus dongi (with S. gui, ‘H’. jii, S. lingyuanensis, and ‘H’. atavismus as junior synonyms), and Morphotype II, corresponding to Huaxiadraco corollatus gen. et comb. nov. (with ‘Huaxiapterus’ benxiensis as a junior synonym). We diagnose each species by compelling and unique combinations of features (including autapomorphies) that are unlikely to be explained by intraspecific variation, as indicated by our qualitative and quantitative analyses. In addition, we regard that premaxillary crest morphology in sinopterines seems to be explained by multiple sources of variation: ontogenetic variation regarding crest presence (with young juveniles being crestless), individual/sexual variation regarding crest development (with mature, crested morphs expressing varying levels of crest size), and interspecific variation regarding crest shape (with a heaped shape in S. dongi and a trapezoidal shape in H. corollatus). We corroborate the view of Sinopterus dongi as being more closely related to the Yixian tapejarid Eopteranodon lii than to Huaxiadraco corollatus, and regard Nemicolopterus crypticus as a very young, undiagnostic, indeterminate sinopterine.

Supplemental Information

Supplemental Information 1 Morphometric dataset.

An Excel file with four sheets: (1) skeletal measurements of the analyzed tapejarid specimens, (2) log-transformed skeletal measurements for the SMA analyses, (3) morphometric dataset of angles and proportions for the clustering analyses, and (4) rostrum deflection angles in tapejarids.

Click here for additional data file.

Supplemental Information 2 Mesquite file.

A nexus-format file for Mesquite, containing the phylogenetic data matrix.

Click here for additional data file.

Supplemental Information 3 TNT file for the phylogenetic analysis.

Click here for additional data file.

We thank the Willi Hennig Society for making TNT freely available. For access to specimens under their care, we thank Cunyu Liu (BPMC), Xuefang Wei (CC-CGS), Caizhi Shen (DNHM), Qiannan Zhang (BMNH), Shaowen Zhang (CAGS), Fangfang Teng (XHPM), Jun Zhang and Honggang Huo (BXGM), Deyu Sun (JPM), Junjie Yao (CDM), Shunxin Jiang and Xiaolin Wang (IVPP), Alexander Kellner, Luciana Carvalho and Uiara Cabral (MN/UFRJ), Dieter Schreiber and Eberhard Frey (SMNK), and Carl Mehling and Mark Norell (AMNH). RVP thanks Tainá Constância (UFABC) and Kamila Bandeira (MN/UFRJ) for help with image software, and Lucas Piazentin (USP) for fruitful discussions. Special thanks to Arthur Brum (MN/UFRJ) for his advice concerning statistical analyses. Special thanks to Cunyu Liu (BPMC) for his great efforts in advancing Liaoning paleontology. We deeply thank reviewers Gabriela Cerqueira, Natalia Jagielska, and Matías Soto, as well as editor Mark Young, for their attentive and helpful comments.

Institutional abbreviations

BMNHC Beijing Museum of Natural History, Beijing, China

BPV Beijing Natural History Museum, Beijing, China

BXGM Benxi Geological Museum, Benxi, China

D Dalian Natural history Museum, Dalian, China

GMN Geological Museum of Nanjing, Nanjing, China

IVPP Institute of Vertebrate Paleontology and Paleoanthropology, Beijing, China

JPM JZMP, Jinzhou Museum of Paleontology, Jinzhou, China

PMOL Paleontological Museum of Liaoning, Liaoning, China

XHPM Xinghai Museum of Prehistoric Life of Dalian, Dalian, China

ZMNH Zhejiang Museum of Natural History, Hangzhou, China

Additional Information and Declarations

Competing Interests

Author Contributions

Data Availability

New Species Registration

The authors declare that they have no competing interests.

Rodrigo V. Pêgas conceived and designed the experiments, performed the experiments, analyzed the data, prepared figures and/or tables, authored or reviewed drafts of the article, and approved the final draft.

Xuanyu Zhou conceived and designed the experiments, performed the experiments, analyzed the data, prepared figures and/or tables, authored or reviewed drafts of the article, and approved the final draft.

Xingsheng Jin conceived and designed the experiments, performed the experiments, analyzed the data, prepared figures and/or tables, authored or reviewed drafts of the article, and approved the final draft.

Kai Wang conceived and designed the experiments, performed the experiments, analyzed the data, prepared figures and/or tables, authored or reviewed drafts of the article, and approved the final draft.

Waisum Ma conceived and designed the experiments, performed the experiments, analyzed the data, prepared figures and/or tables, authored or reviewed drafts of the article, and approved the final draft.

The following information was supplied regarding data availability:

The raw measurements and morphometric dataset ready for analyses, phylogenetic data matrix in Nexus format and in TNT format are available in the Supplemental Files.

The following information was supplied regarding the registration of a newly described species:

Publication LSID: urn:lsid:zoobank.org:pub:E836D564-B986-497A-9E3C-8277EF8EF50E.

Huaxiadraco genus LSID: urn:lsid:zoobank.org:act:39AA06E5-6882-4041-9585-8F2106424C81.

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
