# Peer review of "A taxonomic revision of the Sinopterus complex (Pterosauria, Tapejaridae) from the Early Cretaceous Jehol Biota, with the new genus Huaxiadraco"

_PeerJ, doi:10.7717/peerj.14829_

## Round 0.1 · original submission · Minor Revisions

Dear authors,

Apologies for the delay in returning your manuscript. I had to a get new reviewer as one I was waiting for unfortunately had to drop-out. And I ended up getting three reviewers.

All three agree upon a decision of 'minor revisions'.

I agree with reviewer 3 that a discriminate analysis, or CVA, would be a good next step. Reviewer 2's comment regarding quadrate inclination and post-mortem distortion are important to address as well.

I look forward to receiving your revised manuscript.

·

Basic reporting

No comment.

Experimental design

No comment.

Validity of the findings

No comment.

Additional comments

This paper presents excellents comments and analysis about the tapejarids specimens of China, bringing important information data on the morphological diversity and philogeny of Sinopterinae group. I'm gladly to recommend this paper for publication, although there is some minor modifications indicated in the pdf attached file.

My recommendation to the authors is that they must add some figures of specimens commented in the paper whose did not have the same ones included, as well as figures for specimen comparison, for a better understanding of the paper.

·

Basic reporting

After Naish (2021) skidding on the subject, and numerous descriptive papers, it is a gargantuan task to tackle phylogeny of Jiufotang Tapejarids. With a lot of contradictions and missing data pieces, the paper does a great job summarising it and using sufficiently described, robust methodology acknowledging all data gaps. If the species become synonymised, it'll create the taxa a reference for the study of ontogeny and dynamics in pterosaurs the same way Rhamphorhynchus or Pteranodon or Pteurodaustro now are. The paper is unambiguous, and it uses professional English throughout, although sometimes risks sounding colloquial (albeit that does not need to be strictly addressed).

The paper tackles a multitude of sources, phylogenies and (new + legacy) fossil specimens well; it is easy to navigate and does not become overwhelming and unruly because of the scope of the paper.
The paper however could use some trimming and tightening. It repeats the same points across different segments in the body of the text. Some segments, especially after 1279 - the acknowledging cranial/premaxillary crest with lack of a priori assumption - elements feel like repeating the same points and might need some streamlining. There is some bloating, ie no need to define the abbreviation of pdf (329).

I like the attempt at standardisation in the long descriptive segment of the paper, the set up made it more navigable and easier to compare despite variations in the completeness and quality of fossils. Not all are easily navigated, however, or easily comparable, animal-size diagnostic humeral and cranial lengths which are found in most specimens are sporadically mentioned with dimensions in ontogenetic assessment sections; same with estimated wingspan, the methodology of which is never mentioned (outside of SI morphometric table). A standardised way of obtaining wingspan estimate could be introduced, as it plays into the ontogenetic assessment of the animal in some segments of this paper, and while being just an estimation, it can play into our understanding of this clade. Simple scatter points of humerus/crania lengths (as deployed by Bennett for his synonymization of numerous specimens) could also be useful in visually mapping congregation of sizes (do these correspond to certain lithology or bed that we know of?)
Are there distinctive size groups based on lithology? Do we see the preference of certain sizes or ontogenetic groups, are all represented similarly across two morphotypes, if so why?

There are some descriptions (like the ones on 536) like shallower or deeper, which without numeral support are hard to easily visualise.

The paper is sufficiently illustrated and supported with tables, the pictures are of high quality and the lineart around the fossils is appreciated. The raw data necessary is shared. If I can suggest a figure, it would be an expansion on Figure 18, with full (post-crania included) reconstruction of smallest and largest representatives of both morpho-types; as the current figure focuses only on crania (despite post-cranial features also being diagnostic) of single size-group. This would help to visualise and aid future assessments of two morphotypes.

The paper is self-contained with relevant results to hypotheses. Although, some mention of other Jiufotang pterosaurs could be valuable; as niche partitioning is brought up just once in the text, without pondering on niches currently occupied by other volant animals. Jiufotang is interesting, as it seems to represent numerous representatives of similar istiodactylids, ornithocheiroids and chaoyangopterids, seemingly also differing slightly between species. With similar 1-2 meter wingspans. A short mention of other animals in proximity in a later session of the paper might put this clade into a wider context and explain the need for the large disparity.

Experimental design

The research question is well-defined and soundly explored in the paper. It fills the current knowledge gap with sound analysis on top of describing numerous novel specimens. Methods described are with sufficient detail & information to replicate. I can easily run the Mesquite file and get the same result as figured in the paper.

Some suggestions:

Avoid using colour as a proxy for sedimentology (160-161). More citations on the lithology section are needed (160-166), along with some information on palaeoenvironment and pterosaur assemblages in the region for better contextualization of the assemblage, and controls of preservation which all feed into the true representation of biodiversity of Jiufotang. Because of the lack of detailed provenance details for every fossil and I appreciate the attempt at the correlation (as displayed in Figure 1). Lithology can tell us a lot about variation in size and ontogeny - this has been used to ascertain ontogenetic phases by Bennett (1995) in other mass assemblages that too led to confluence of numerous species to single species in Solnhofen - and tried to describe patterns in preservation down to seasonality and mass mortality. Taphonomic factors are not required but given varied influence on specimens (crushing, disarticulation), mention of dynamics of preservation in the formation might help with assessing confidence in reading features of the animals preserved.

While I like the approach and agree with the conclusion of the two Morphotypes. One of the characters, quadrate inclination, operates at a small angle of 10o difference. As specimens are crushed, disarticulated, unfused and usually preserved at slightly differing angles - how can you be certain that the small difference is not down to a fluke and a solid character? Define how you standardised the angle measurement to improve reproducibility.

I am glad the Nemicolopterus crypticus is noted and has its section to it with sound reasoning of avoidance of genus specification. Albeit, it would be interesting to see where it would plot, or how much would it influence the PCA/SMA.

Remember that ontogenetic cranial fusion can show ossification plasticity and be representative of cranial kinesis. Individuals that appear large and fused might be osteologically immature too.

Validity of the findings

This paper is in a good state; the methodology is sound and well described; selected autapomorphies are valid - although shame more specimens currently held in numerous Chinese institutions cannot be assessed to fully aid this investigation, but this means the paper encourages replication and reference.

I am glad the paper acknowledges the limitations of the primary data source and keeps these apparent throughout the paper. All underlying data is provided and sound, acknowledging its limitations. The attached supplementary files are easy to navigate and run. The TNT file runs and reflects results seen in figures and are addressed in the paper; Mesquite and excel datasheets are easy to navigate and readable, from measurements to SMA analyses.

Conclusions link to the original research question, although the abstract could mention the non-crest-related characters and key diagnostic characters between morphotypes.

·

Basic reporting

No comment

Experimental design

No comment

Validity of the findings

No comment

Additional comments

First of all, my congratulations to the authors for this well-written, rigurous article that finally solves a complet taxonomic history regarding Jehol tapejarids. It is hard to find any way to improve this article. Minor suggestions below

Lines 58 to 65 – “further” appears several times. Use synonyms such as “additional”
Line 94 – “there are more likely”
Line 129 – perhaps “dangerous” is not the most appropiate adjective
Lines 162 to 163 – all rock types are given in singular, but conglomerates is in plural
Line 165 – “through a parallel unconformity”
Line 230 – “among which”
Line 244 – delete “)”
Line 283 – “pruned”?
Line 285 – “principal”
Linea 636 – “crest”
Linea 653 – synonym for unclear to avoid repetition in same line
Line 768 – rephrase
Line 859 – delete dash in well-preserved
Line 928 – “in posterior view” rather than “from the posterior view”
Line 939 – despite being relatively complete
Line 941/942 – “still” is repeated
Line 1143 – replace surrounds for a more proper term
Line 1300 – “still” is repeated

Multivariate analyses: have the authors considered making also a discriminant analysis to see if the two morphotypes are still being recovered (I guess yes, given the clear separation in the PCA) and which variables allow for better discrimination among morphotypes?

Phylogenetic analyses section: I think it would be easier to the reader if the authors rephrase the character states in the text. For example: “concave dorsal margin of the mandibular ramus”. Mentioning some support values (Bremer, Jacknife and/or Bootstrap) may be useful. Last paragraph of this section is a nice discussion.

Figure 18. Very helpful figure summarizing relevant variation!

Figure 21 caption. I suppose different symbols in the tree indicated node-based and stem-based clades. Please mention it in the caption.


Figure 22. Superb artwork!

Last but not least, about the election of the genus name Huaxiadraco. I suppose the authors discarded the possibility of confussion with the genus Huaxiapterus (despite being no longer valid) found in the same unit and horizons.

---

## Round 0.2 · Minor Revisions

Dear authors,

Many thanks for your revised manuscript. Based on your tracked changed text, and the response to reviewers comments I have decided it does not need to go back out to review.

There are four minor things I'd like to bring up:

1. typos in the newly added text (abundand instead of abundant in the abstract). 2. what is the alpha-value you're using for the KW tests? Based on table 3 I'm assuming you're using 0.05? Given that you're making multiple comparisons within a sample would it not be necessary to correct the alpha value? A Bonferroni correction might be too strong in this case, but another method might be suitable.

3. the consistency index and retention index. Technically you are reporting the ensemble consistency index and the ensemble retention index.

4. in Table 1, englobes? Would encompasses or contains not work better?

One other issue, the new PhyloCode-compliant nomina. In Table 1 Caupedactylia and Eutapejaria are listed as being established in your manuscript ("This work"). But neither are defined or discussed in the text. Can you add the relevant sections to the text, much like they appear in the PhyloNoms book?

Eutapejaria doesn't appear in Figure 21. Can it be added?

---

## Round 0.3 · accepted · Accept

Dear authors,

Thank you for your quick response to my previous queries. I'm happy to inform you that your manuscript has been accepted for publication.

The production team will be in contact with you shortly to take you through the proofing stages.

Once again, congratulations and I hope you will choose PeerJ as your publication venue in the future.